# MS2Query: reliable and scalable MS² mass spectra-based analogue search

Niek F. de Jonge [1] ✉, Joris J. R. Louwen[1], Elena Chekmeneva [2],
Stephane Camuzeaux [2], Femke J. Vermeir[3], Robert S. Jansen [3],
Florian Huber [4,6] ✉ & Justin J. J. van der Hooft [1,5,6] ✉

Metabolomics-driven discoveries of biological samples remain hampered by the grand challenge of metabolite annotation and identification. Only few metabolites have an annotated spectrum in spectral libraries; hence, searching only for exact library matches generally returns a few hits. An attractive alternative is searching for so-called analogues as a starting point for structural annotations; analogues are library molecules which are not exact matches but display a high chemical similarity. However, current analogue search implementations are not yet very reliable and relatively slow. Here, we present MS2Query, a machine learning-based tool that integrates mass spectral embedding-based chemical similarity predictors (Spec2Vec and MS2Deep-score) as well as detected precursor masses to rank potential analogues and exact matches. Benchmarking MS2Query on reference mass spectra and experimental case studies demonstrate improved reliability and scalability. Thereby, MS2Query offers exciting opportunities to further increase the annotation rate of metabolomics profiles of complex metabolite mixtures and to discover new biology.

Wide-screen untargeted metabolomics applications are increasingly used to understand complex metabolite mixtures. To boost the metabolite structure annotation rate, mass spectrometry fragmentation approaches are a key source of information in the field of metabolomics[1]. Many improvements have been made in automatically elucidating molecular structure from mass spectrometry fragmentation spectra (also referred to as MS/MS or MS² spectra)[2]. However, it remains very challenging to reliably determine structures based on MS² spectra[3]. Currently, three main types of approaches to determine molecular structures from MS² spectra exist: matching against annotated mass spectral library spectra[4–9], by using fragmentation trees[10–12], or by predicting mass fragmentation spectra from chemical structures to match against molecular structure databases[13–18]. However, all these

approaches still have important limitations. Many of these methods were recently reviewed by our group, in particular those using machine learning[19].

One inherent limiting factor of mass spectral library matching is that annotated spectra for only a fraction of the chemical space are known. For example, the GNPS[20] public mass spectral libraries contain about 2.5% of known natural products[21]. When searching for exact matches, this typically results in finding a few exact spectral matches (with corresponding molecular masses) in a given sample[22]. To overcome this limitation, several methods try to search larger structural databases like Pubchem[23] for potential matches. These methods typically rely on first predicting spectra from structures by using in silico fragmentation, followed by comparing MS² spectra to these predicted

[1]Bioinformatics Group, Wageningen University & Research, 6708 PB Wageningen, the Netherlands. [2]National Phenome Centre, Section of Bioanalytical Chemistry, Division of Systems Medicine, Department of Metabolism, Digestion and Reproduction, Faculty of Medicine, Imperial College London, Hammersmith Hospital Campus, London W12 0NN, UK. [3]Department of Microbiology, Radboud Institute for Biological and Environmental Sciences, Radboud University, 6525ED Nijmegen, the Netherlands. [4]Centre for Digitalization and Digitality (ZDD), University of Applied Sciences Düsseldorf, Düsseldorf, Germany. [5]Department of Biochemistry, University of Johannesburg, Auckland Park, Johannesburg 2006, South Africa. [6]These authors jointly supervised this work: Florian Huber, Justin J.J. van der Hooft. ✉e-mail: niek.dejonge@wur.nl; florian.huber@hs-duesseldorf.de; justin.vanderhooft@wur.nl

spectra[13–15]. Even though these methods are promising, they are still far from perfect at predicting in silico fragmentation, especially for larger molecules such as complex specialised metabolites or lipid-like molecules. Other methods try to retrieve information directly from the MS² spectra without relying on spectral library databases by creating fragmentation trees. Fragmentation trees have been used to predict molecular formulas[10], to match against structural databases[10,12], to predict molecular fingerprints[24] and recently to predict completely novel structures from MS² spectra[11]. These methods show excellent results for smaller metabolites of <400 Da; however, for larger metabolites these approaches are still not fully reliable in returning correct elemental formulas and candidate structures. Besides that, the computation time to determine the fragmentation trees also increases substantially[19]. Natural mixtures typically contain considerable amounts of larger metabolites (>800 Da), and this thus poses challenges on the mass spectral interpretation.

A different approach to increase the percentage of spectra for which chemical information can be retrieved is by searching for analogues instead of exact matches[4,9,25–27]. This approach also relies on annotated mass spectral libraries but aims at finding chemically similar molecules, without the need for them to be identical. To perform a succesful analogue search, it is important to have a spectral similarity score that serves as a good proxy for chemical similarity even if two molecules are not identical. A first improvement made in this direction was the development of the modified cosine score, which in contrast to the cosine score also uses neutral losses for determining spectral similarity[4,28], see Supplementary Note 8 for more details. This makes the modified cosine score less sensitive to a small chemical modification. However, multiple small chemical modifications can still result in a large decrease in mass spectral similarity based on the modified cosine score, which limits its ability to serve as a proxy for chemical similarity[19,29–31]. Recently, two machine learning-based methods were developed that outperform cosine-based scores in predicting chemical similarities from MS² mass spectral pairs; the unsupervised Spec2Vec[30] and the supervised MS2Deepscore[32]. We hypothesised that their chemical similarity predictions offer great potential for performing a reliable analogue search.

Current implementations of an analogue search only consider one library spectrum to predict chemical similarity. However, for a good analogue other chemically closely related library structures are expected to have similar structures to a query spectrum as well. MS2Query uses this principle to improve prediction quality of an analogue search, by additionally using MS2Deepscore for similar library structures to predict if a molecule is a good analogue. In addition, MS2Query combines the strength of both MS2Deepscore

and Spec2Vec and uses precursor m/z to further improve prediction quality.

Here we present MS2Query, a tool for rapid large-scale MS² library matching that enables searching both for analogues and exact matches in one run. MS2Query can reliably predict good analogues as well as exact library matches. We demonstrate that MS2Query is able to find reliable analogues for 35% of the mass spectra during benchmarking with an average Tanimoto score of 0.63 (chemical similarity). This is a substantial improvement compared to the modified cosine score-based method, which on the same test set resulted in an average Tanimoto score of 0.45 with settings that resulted in a recall of 35% (percentage of query spectra for which a match is predicted). To create the used benchmarking test set, any exact library matches were removed from the reference library to make sure the best possible match that can be found is an analogue. Besides thorough benchmarking on annotated library spectra, MS2Query was also used for multiple case studies. The higher quality of predictions by MS2Query offers exciting opportunities to further increase the annotation rate of metabolomics profiles from complex metabolite mixtures and to discover new biology. MS2Query is available as a well-tested and open-source Python library that facilitates easy access to both researchers and developers.

## Results

### MS2Query combines several machine-learning approaches

The workflow for running MS2Query first uses MS2Deepscore[32] to calculate spectral similarity scores between all library spectra and a query spectrum (Fig. 1). In contrast to existing methods, no preselection on precursor m/z is needed. By using pre-computed MS2Deepscore embeddings for library spectra, this full-library comparison can be computed much faster than existing alternatives (see Speed Performance section). Next, the top 2000 spectra with the highest MS2Deepscore are selected. MS2Query optimises re-ranking of the best analogue or exact match at the top by using a random forest that combines five features. The random forest predicts a score between 0 and 1 between each library and query mass spectrum. By using a minimum threshold for this score, unreliable matches can be filtered out.

As input for the random forest model, MS2Query uses five different features, calculated between the query spectrum and each of the 2000 preselected library spectra. These features are Spec2Vec similarity score[30], query precursor m/z, precursor m/z difference, a weighted average MS2Deepscore over 10 chemically similar library molecules, and the average Tanimoto score for these 10 chemically similar library molecules. The random forest model was trained to

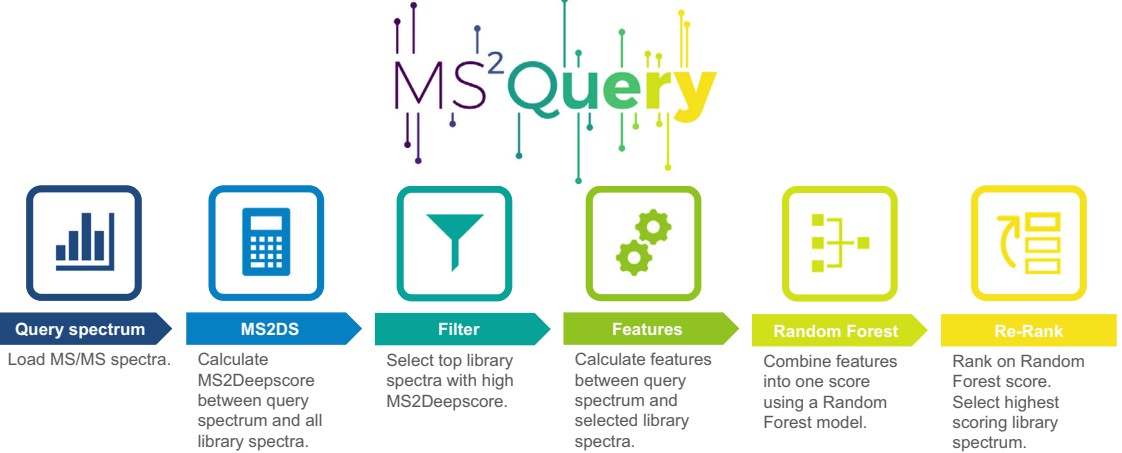

**Fig. 1 | Schematic workflow of MS2Query.** MS2Query searches for both exact matches and analogues in a reference library. First, potential candidates are selected based on MS2Deepscore, followed by re-ranking the spectra by using a random forest model.

predict Tanimoto scores (molecular fingerprint-based chemical similarity) based on these 5 features. More details about the rationale behind these features can be found in Supplementary Note 2 and the Methods section.

The feature that has the biggest impact on the increased performance is the *Average MS2Deepscore of multiple library molecules*, see Supplementary Note 2. This feature builds on the following principle: if two library molecules are chemically very similar, it is expected that if one of these library molecules is a good analogue to the query molecule, the other is a good analogue as well. For this reason it is expected that the MS2Deepscore between the spectrum of a chemically similar library molecule and the query spectrum is also high in case of a good analogue. MS2Query is the first mass spectral library searching method that uses this principle to re-rank candidate molecules.

## Speed performance

Running MS2Query on 5987 test spectra took 1 hour and 14 minutes (80 spectra per minute) on a normal laptop with an 11th generation Intel Core i5-1135G7 and 16 GB of RAM using version 0.3.2 of MS2Query. The test spectra were matched against a library of 302,514 spectra, without doing any preselection on the precursor m/z difference. An analogue search on the same test set using the Modified cosine score and a preselection on a maximum precursor m/z difference of 100 Da took 9 hours and 24 minutes (10,6 spectra per minute). Note that this would take much longer with a larger maximum precursor m/z difference. For doing the modified cosine score calculations the implementation of matchms[33] was used, which is optimized for performance.

## MS2Query has an improved performance in benchmarking

To test the performance of MS2Query, models were trained using publicly available mass spectra from GNPS. These spectra were first cleaned and filtered, including unifying the format of the metadata, filtering out spectra with less than three peaks and normalizing intensities.

The performance on finding exact matches and finding analogues was tested separately using two different test sets. The test set for searching for exact matches ('exact matches test set') contains spectra that have at least one spectrum in the library from exactly the same molecule. The test set to test the performance for an analogue search ('analogues test set') contains spectra that do not have an exact match to a library spectrum. Thus, for this test set, the best possible match has to be an analogue of the query spectrum. To create the analogues test set, 20-fold cross-validation was performed and the data split was done randomly on all unique 2D structures. 20-fold cross-validation was chosen to ensure that a large-enough training set was used to not compromise on overall model performance. In case of the exact matches test set a test spectrum was randomly selected for each unique 2D structure with at least 2 available spectra. 20 test sets were created by randomly selecting unique test spectra. None of the testing spectra were used for training MS2Deepscore, Spec2Vec and MS2Query to ensure that there is no data leakage between the models. In case of the exact matches test set, the spectra are spectrum-disjoint, meaning that no spectrum in the test set was used for training. The analogue search test set is structure-disjoint, meaning that there were no spectra in the training set that correspond to any of the 2D structures in the test set. 2D structures were used, since tandem mass spectrometry cannot discriminate between different stereoisomers, since they yield similar or identical fragmentation mass spectra.

The performance of MS2Query was compared to MS2Deepscore, the cosine score and the modified cosine score. As a metric for the quality of a predicted analogue the average Tanimoto score between the test molecules and predicted analogues is used. The Tanimoto score[34] is a metric for chemical similarity between two

molecules, based on chemical fingerprints[35]. For all methods a minimal threshold can be used to vary the percentage of query spectra for which a match is predicted (recall). The quality of predictions increases with more stringent thresholds for all methods, but the recall decreases. To assess the performance of an analogue search, the recall is compared to the average Tanimoto score on the 'analogues test set' (Fig. 2a). Across all recall values, MS2Query predicts analogues of better quality than comparable search methods relying solely on MS2Deepscore or on the modified cosine score. At a high recall, the observed increase in performance is smaller, which suggests that the main added value of MS2Query is a better removal of bad matches as compared to the other methods. This demonstrates the importance of using a sufficiently high threshold for the MS2Query score.

To determine the performance for finding an exact match, the percentage of predictions that is an exact match for the test spectra is calculated for the 'exact matches test set' (Fig. 2a). The preselection on precursor m/z difference was set to 0.25 Da for MS2DeepScore and the cosine score, while for MS2Query no pre-filtering on the mass difference was used, since MS2Query used the exact same settings and model as for the analogue search. Figure 2b shows that MS2Query performs better at finding exact matches compared to search methods relying on MS2Deepscore or the cosine score.

Additional analysis of performance for different mass ranges can be found in Supplementary Note 4. Supplementary Note 7 shows the benchmarking results of the analogue test set without any preselection on precursor m/z for the reference methods.

## Case studies on experimental datasets of complex metabolite mixtures

MS2Query was run on four case studies, to demonstrate that MS2Query also performs well on newly generated experimental data. Mass spectra obtained using different LC-MS/MS assays for a urine sample, two blood plasma samples, and an anammox bacterial sample set were analysed using MS2Query and GNPS analogue search. The results of the case studies were manually validated and partially confirmed by in-house reference standards. Though informative, we would like to stress that a fair comparison of the performance in these case studies is challenging, since often no ground truth can be found for all spectra and judging whether two chemical structures are analogues remains to some extent subjective. The detailed results for all case studies can be found in the Supplementary Data 1. Below we highlight some of the results of four case studies to illustrate that MS2Query is able to predict useful exact matches and analogues for newly generated data.

Figure 3a shows the number of spectra for which MS2Query predicted a match (recall) for the four case studies. The recall for the four case studies is highly variable, but on average, the case studies do not have a clear higher or lower recall compared to the benchmarking test set used. Figure 3b shows that the ratio between the number of predicted analogues (mass difference >1 Da) and predicted exact matches (mass difference <1 Da) differs between the case studies. Manual validation shows that most predictions by MS2Query were analogues or exact matches that matched with prior biochemical knowledge on the sample (Fig. 3c). This confirms that MS2Query is able to generate relevant predictions for newly generated experimental data.

The NIST plasma sample analysed by lipid profiling assay in positive ionization mode contained 139 spectra for which MS2Query predicted 75 matches. Since this blood plasma sample was analysed by an LC-MS assay tailored for the profiling of lipids, the resulting MS2Query predictions were, as expected, mainly lipids. 72 out of 75 matches predicted by MS2Query were lipids. This indicates that MS2Query is able to reliably find analogues which consistently match the correct compound class.

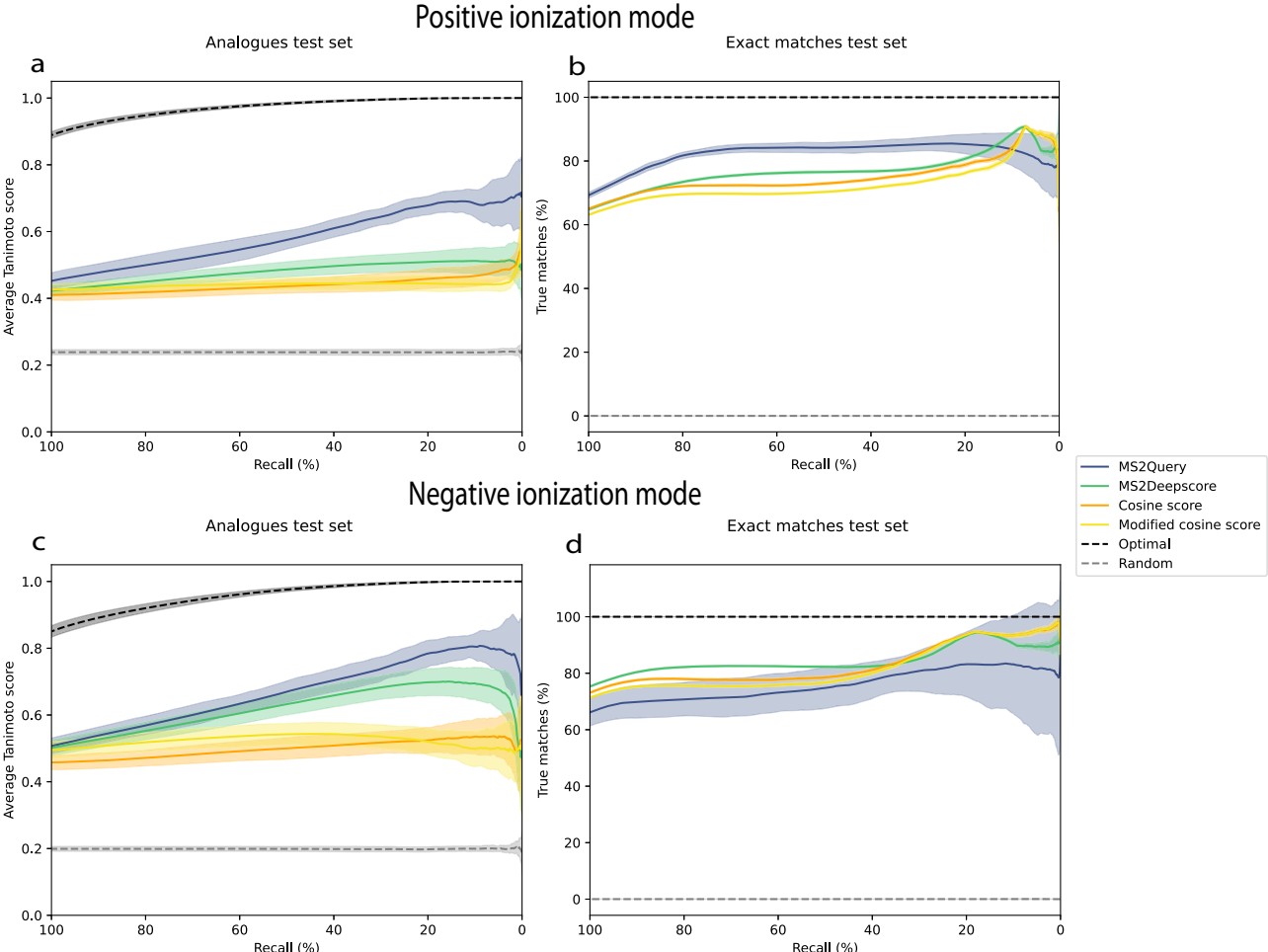

**Fig. 2 | MS2Query benchmarking results.** MS2Query is more accurate for finding analogues than using MS2Deepscore or modified cosine score and is more accurate at predicting exact matches in positive mode at high recall than using MS2Deepscore, the cosine score or the modified cosine score. The threshold for MS2Query, MS2Deepscore, cosine and modified cosine is varied, resulting in different recalls. The random results show the results if random matches would be selected and the optimal results show the performance if the best structural match in the library was selected. Results of 20-fold cross-validation are shown. The mean of these 20 test sets are shown and the standard deviation is highlighted. Source data are provided as a Source Data file. **a** The 'analogues test set' is used with spectra that have no exact match in the library, therefore the best possible match is always an analogue. For MS2Deepscore, cosine score and modified cosine score, library spectra are first filtered on a mass difference of 100 Da. The relationship between recall and average Tanimoto score (chemical similarity) is plotted. For each threshold the average over the Tanimoto scores between the correct molecular structure and the predicted analogues is calculated. **b** The 'exact matches test set' is used, all these test spectra have at least 1 exact structural match in the reference library. For MS2Deepscore and modified cosine score, library spectra are first filtered on a mass difference of 0.25 Da, while MS2Query does not use any pre-filtering on mass difference, and uses the exact same settings as for the analogue search. The percentage of true positives is given. A match is marked as true positive if the 2D structure is correct. **c** The same plot as Fig. 2a, but for a model trained on spectra in negative ionization mode. **d** The same plot as Fig. 2b, but for a model trained on spectra in negative ionization mode.

## Discussion

Structural elucidation based on mass spectrometry fragmentation data remains hampered by a limited number of reference mass spectra in spectral libraries. Only a fraction of the mass spectra in experimental data can therefore be annotated. Many different approaches target this structural annotation problem, for instance fragmentation tree based methods[10–12], or approaches generating in silico spectra based on structural libraries[13,14]. Even though these are promising approaches, the problem of automatically assigning structures to mass spectra remains unsolved. Searching for so-called analogues is an attractive alternative to exact library matching. Analogues are library molecules, which are not exact matches but are structurally very similar. Analogues can be used as a starting point for complete annotation, to select metabolites of interest, or for direct biological interpretation. A benefit of searching for analogues compared to compound class prediction is that analogues make the biochemical interpretation more flexible. The choice is not limited to specific

chemical compound classes but can be extended to specific side groups for metabolites of interest, involvement in certain pathways, or relatedness to specific drugs or contaminants. Furthermore, searching for analogues can potentially help in efficiently increasing the chemical diversity of public libraries. If an analogue search does not return any matches, this metabolite is likely to be unrelated to known metabolites. Prioritizing such metabolites for structural identification by NMR spectroscopy would be an efficient way to increase the chemical diversity of public libraries. Here, we introduce MS2Query, a tool that can search a large mass spectral library both for exact matches and analogues. Based on the performed benchmarking, we expect that searching for analogues in currently publicly available mass spectral libraries, MS2Query will typically result in useful analogues for about one third of all molecules present in a complex sample. The precise fraction, however, will vary depending on the exact composition and origin of a sample and the similarity of its molecules with those in mass spectral libraries.

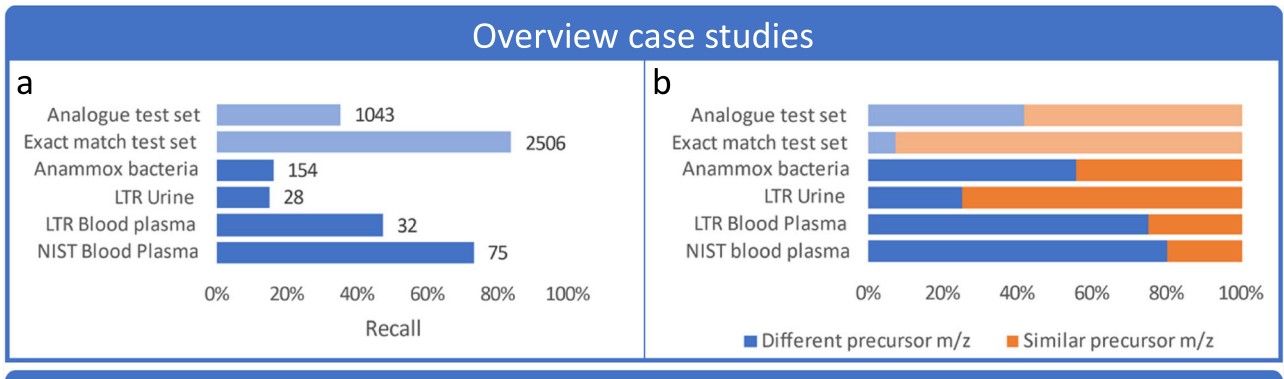

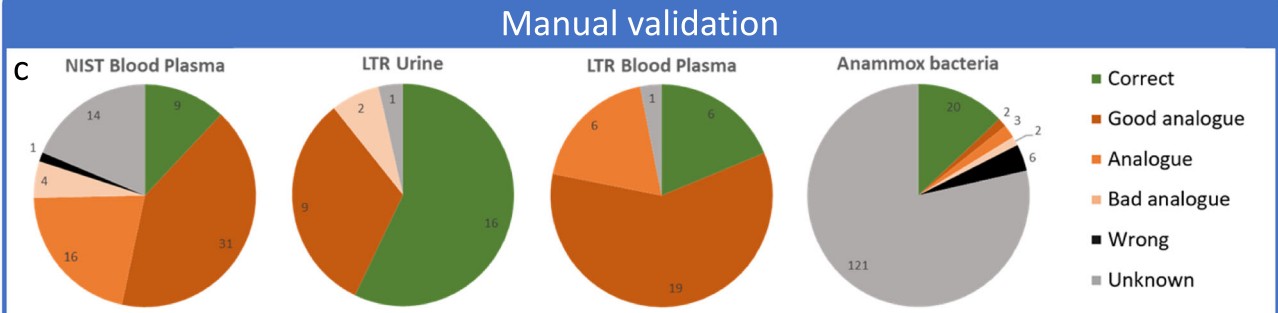

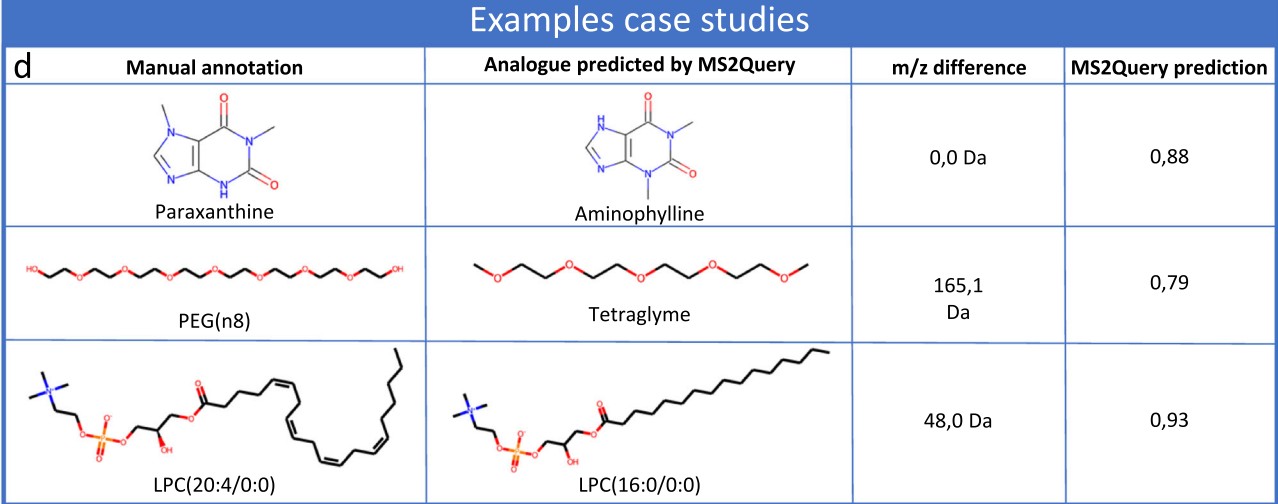

**Fig. 3 | Highlights of the results of the case studies.** The same MS2Query model was used for all test sets, for more details about the model used for the case studies, see Supplementary Note 1. A minimal threshold of 0.633 for the random forest score was used to determine if an analogue was selected. The threshold of 0.633 was selected, since this resulted in a recall of 35% for the "analogue test set". Source data are provided as a Source Data file. **a** The variation of recall across case studies using the same settings. **b** The percentage of query spectra with a predicted analogue (precursor m/z > 1 Da) is compared to the percentage of spectra with an exact match predicted (precursor m/z < 1 Da) **c** Results were manually validated based on the retention time MS1 mass and MS2 spectra, by comparing to online libraries or in-house reference standards. These reference standards were used to judge the quality of the predicted analogues. In the Supplementary Note 6 more details about the validation can be found. For the anammox bacteria sample set, tentative validation was attempted for 50 features. **d** Three examples of predictions for mass spectra in the case studies. These examples came from the case study test sets LTR Urine, LTR Blood Plasma, and NIST Blood Plasma in that order. For LPC(20:4/0:0) the exact position of the double bonds could not be determined and was therefore guessed for the visualization.

Comparison with cosine score, modified cosine score, and MS2Deepscore shows that MS2Query performs better both at finding exact matches as well as finding analogues for positive mode MS$^2$ spectra. Using a modified cosine score is a common approach for doing an analogue search, for instance implemented on GNPS[4] and MASST[9]. Even though we demonstrate that MS2Query is able to rapidly provide reliable analogues for unknown substances, there is still room for improvement. The current version was trained using available data from GNPS[20]. While a very valuable resource, we do expect that our models will notably improve when our library is built from larger and chemically more diverse datasets. For negative mode mass spectra, MS2Query performed worse, which is probably due to the lower number of publicly available mass spectra in negative mode and the fact that negative mode mass spectra contain less mass fragments compared to positive mode mass spectra. Nevertheless, MS2Query currently represents a substantial step forward in reliability, thereby creating opportunities to use analogues to get more reliable insights into unknown mass spectra.

In the four case studies the recall varies from 15 to 75% with the same settings (Fig. 3). The observed variation can be due to differences in the quality of the acquired spectra, the masses of the metabolites, or the differing similarity between the metabolites in the sample and the metabolites in the reference libraries. This, in combination with the challenges of manually validating results, makes it hard to objectively

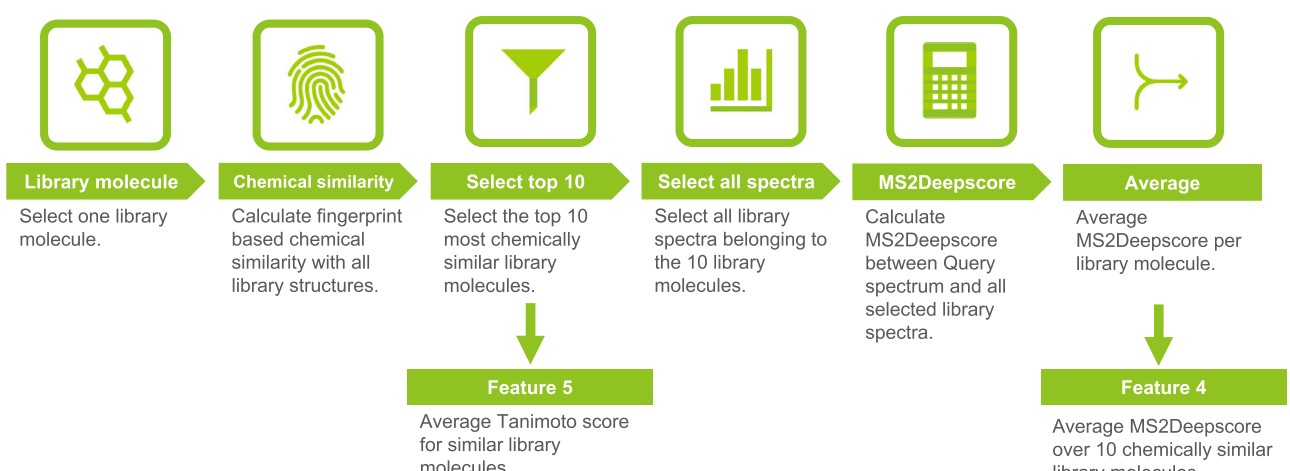

**Fig. 4 | Workflow for calculating two input features of the random forest model.** Feature 5 is the Average Tanimoto score for similar library molecules and feature 4 is the average MS2Deepscore over 10 chemically similar library molecules.

judge if MS2Query performs similarly on newly generated data, compared to the benchmarking test set. Nonetheless, the case studies show that MS2Query can generate useful results for newly generated experimental data and that it can contribute to new biochemical insights based on previously unconnected analogues.

Since a preselection on MS2Deepscore is the start of our method, the improved performance of MS2Query compared to MS2Deepscore shows the added value of using the five features and the random forest for re-ranking the library spectra. Additional analysis of the feature importance indicates that each of the five features used contain relevant information for correctly ranking candidate structures (Supplementary Table 1 and 2). The most important feature for the performance of MS2Query is using the average MS2Deepscore of multiple library structures; this shows the value of using multiple library spectra for predicting good analogues. Besides the five used features, multiple other features were tested as well, for instance the cosine and modified cosine score. These other features were not selected, since they did not improve the performance of the model. Details about the other features that were tested can be found in the Supplementary Note 3.

MS2Query is available as an easily installable python package, which is stable and well-tested. The model as well as the library mass spectra used are available on Zenodo. MS2Query is fully automatic and was designed with the end-user in mind. For example, it outputs a CSV file with all relevant information about the found matches for the query spectra. For each found analogue it also returns the chemical compound classes based on ClassyFire[36] annotations to facilitate biochemical interpretation of the results. MS2Query is optimized for speed and working memory usage, which makes it possible to run MS2Query on a normal laptop on 1000 spectra within 13 min against a reference library of 302 514 spectra, without doing any preselection on precursor m/z difference. The scalability of MS2Query is an encouraging step toward higher-throughput large-scale untargeted metabolomics workflows, thereby creating the opportunity to develop large-scale full sample comparisons.

## Methods
### Workflow MS2Query
MS2Query builds on the improvements of two machine learning-based methods, developed to predict chemical similarity from $MS^2$ mass spectral pairs; Spec2Vec[30] and MS2Deepscore[32]. These methods perform especially well at predicting chemical similarity for molecules that are similar but are chemically not exact matches. This makes Spec2Vec and MS2Deepscore very suitable for an analogue search.

The workflow for running MS2Query first uses MS2Deepscore to calculate spectral similarity scores between all library spectra and a query spectrum. The top 2000 spectra with the highest MS2Deepscore are selected. To optimally rank these 2000 spectra, MS2Query calculates 5 features which are combined by a random forest model. The prediction of the random forest model is used to rank the 2000 preselected library spectra (See Fig. 1). As input for the random forest model, MS2Query uses 5 different features, calculated between the query spectrum and each of the 2000 preselected library spectra. The features are 1. Spec2Vec similarity, 2. query precursor m/z, 3. precursor m/z difference, 4. an average MS2Deepscore over 10 chemically similar library molecules, and 5. the average Tanimoto score for these 10 chemically most similar library molecules.

The Average MS2Deepscore of multiple library molecules (feature 4), builds on the following principle. For two library molecules that are chemically very similar, it is expected that if one of these library molecules is a good analogue to the query spectra, the other is a good analogue as well. For this reason it is expected that for a good analogue the MS2Deepscore between such a chemically similar library molecule and the query spectrum is also high. This is captured in this feature by calculating the average MS2Deepscore between a query spectrum and all spectra of 10 chemical similar library molecules (Fig. 4). These 10 library molecules are selected based on the known chemical structures of the spectra in the library, by selecting the library structures with the highest Tanimoto score. For each of the 10 library molecules all corresponding library spectra are selected. As an input feature for the random forest model, the average over the MS2Deepscore for the 10 library structures is used (Feature 4). In addition, the average of the Tanimoto score between the starting library structure and the 10 library structures is used as an additional input feature (Feature 5). Multiple variations of the implementation of this feature were tested and the best performing implementation was selected. These other implementations used weighting based on the Tanimoto score, or weighting the MS2Deepscore for each spectrum equally instead of using the average MS2Deepscore per library structure. These other implementations and their performance are described in more detail in Supplementary Note 3.

### Tanimoto scores as structural similarity label
First, an rdkit[35] daylight fingerprint (2,048 bits) is generated from the SMILES for each unique 2D structure in the library. Unique 2D-structures were selected by selecting the first 14-characters of the InChIKeys in the library. The SMILES were first sanitized by rdkit. If multiple spectra with the same InChIKey exist in the dataset, a

spectrum with the most frequently occurring InChI was selected and was used for all spectra with the corresponding InChIKey. A Tanimoto score[34] was calculated between the molecular fingerprints for each pair of InChIKeys. The Tanimoto score is used as an indication for structural similarity of that pair. These Tanimoto scores are used as labels for training MS2Deepscore and MS2Query and for selecting chemically similar library molecules to calculate an average of the MS2Deepscore of multiple chemically similar library spectra.

### Data cleaning

For training and testing of MS2Query, we used data from GNPS. For the k-fold cross-validation the spectra were downloaded from (https://gnps-external.ucsd.edu/gnpslibrary/ALL_GNPS_NO_PROPOGATED.mgf) on the 1st of November 2022. For the case studies and the determining of the feature importance the GNPS dataset used was downloaded from GNPS (https://gnps-external.ucsd.edu/gnpslibrary/ALL_GNPS.mgf) on the 15th of November 2021, 20:00 CET. More details about the model used for benchmarking the case studies can be found in Supplementary Note 1.

The dataset was first cleaned using matchms[33]. The metadata was cleaned to get a uniform format and to remove or correct misplaced metadata. The dataset is split into positive and negative mode spectra. The intensities of the mass fragmentation peaks are normalised. Peaks above 1000 Da were removed and peaks with an intensity of less than 0,1 % of the highest peak were removed. For spectra with more than 500 peaks, the peaks with the lowest intensities were removed. Spectra with less than 3 peaks were completely removed from the library. Some spectra in the GNPS library do not have an InChIKey stored. A method from matchms extras was used to add missing InChIKeys by searching the compound name and molecular formula on PubChem. The library spectra were split into annotated and unannotated spectra. A spectrum was considered fully annotated if it has a valid SMILES, InChIKey and Inchi. The unannotated spectra were used as additional training spectra for Spec2Vec, since Spec2Vec is unsupervised. The unannotated spectra were not used for training MS2Deepscore, MS2Query or for the test spectra.

### Training models for MS2Query

MS2Query uses MS2Deepscore and Spec2Vec models, for all benchmarking new models for MS2Deepscore and Spec2Vec were trained to ensure that none of the test spectra were used for training these models (Fig. 5). MS2Deepscore was trained on all fully annotated spectra from the GNPS library, using the same settings as used for the MS2Deepscore publication[32]. A Spec2Vec model is trained using all spectra from the GNPS library, both annotated and unannotated spectra. The model is trained in 30 epochs using binning on 2 decimals[30].

The random forest model used by MS2Query was trained on pairs of annotated spectra using five different features to predict the Tanimoto score between the two structures of each pair. To generate the

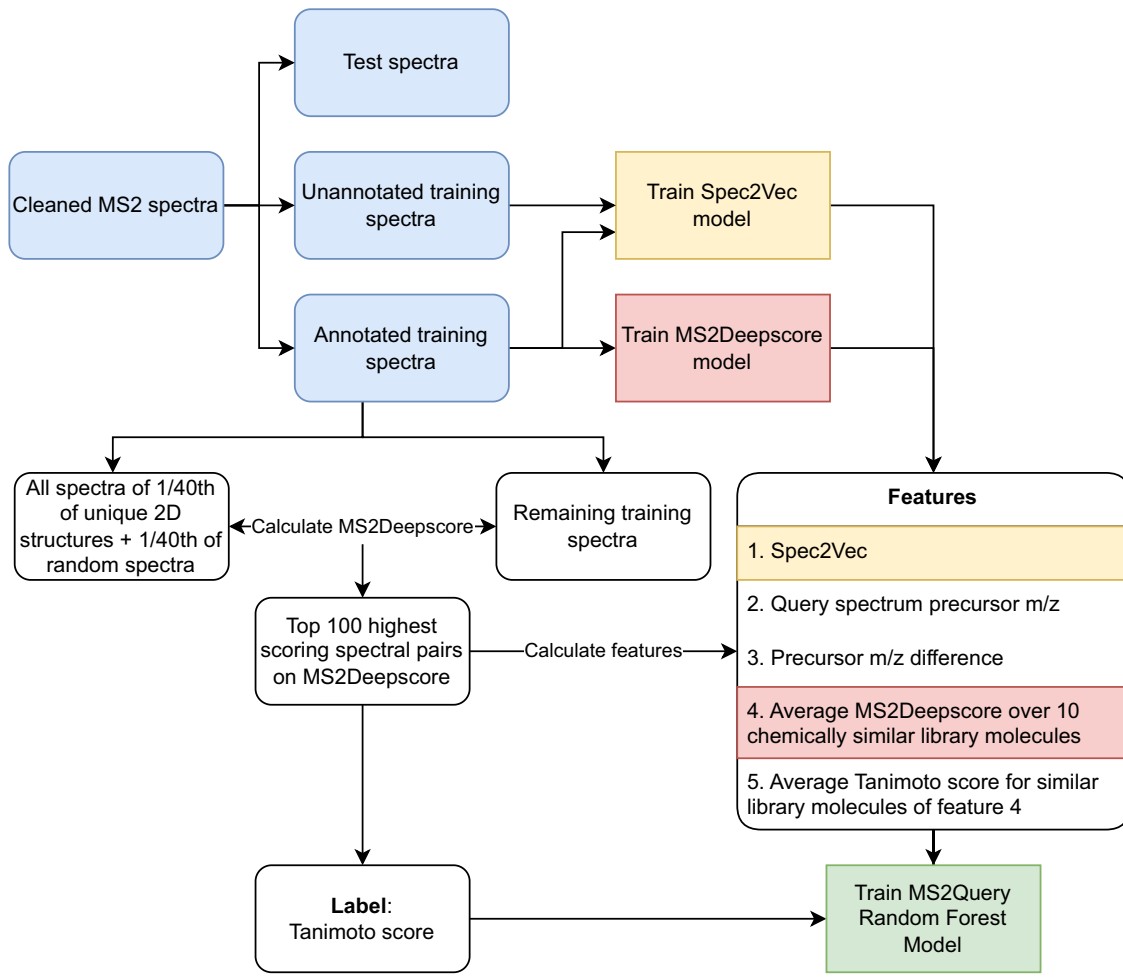

**Fig. 5 | Workflow for MS2Query model training.** Workflow for training the MS2Deepscore model, the Spec2Vec model and the random forest model used by MS2Query. Rounded boxes indicate mass spectral handling steps, whereas squared boxes are indicating machine learning model training steps. The blue colour highlights preparation steps of the mass spectral data prior to model training, the yellow colour the Spec2Vec model, the red colour the MS2DeepScore model, and the green colour the MS2Query model.

training spectrum pairs all library spectra of 1/40th of unique InChIKeys and 1/40th of the remaining spectra were randomly selected from the library spectra. The spectrum pairs were generated by starting with one spectrum from this set and creating spectrum pairs with the 100 library spectra that have the highest scoring MS2Deepscore for this spectrum. More details about the motivation for selecting the top 100 highest scoring spectra for training can be found in Supplementary Note 5.

The implementation of scikit-learn[37] was used for the random forest model. The mean squared error was used as a loss function. The number of estimators was set to 250 and the max depth to 5. The implementation of scikit-learn was used to calculate the feature importance of the 5 scores used. This method is based on an impurity-based feature importance, also known as the Gini importance[38].

Beside these 5 features, multiple other features were tested as well, for instance the cosine and modified cosine score. These other features were not selected since these did not improve the performance of the model. Details about the other features tested can be found in Supplementary Note 3.

### Benchmarking
MS2Query was designed to search for analogues and exact matches in one run, two different types of test sets were generated to test the performance on these two goals, an "analogue test set" and an "exact matches test set". Benchmarking for the analogue test set was done using 20-fold cross-validation. Meaning that the training dataset was split into 20 test sets. The MS2Deepscore, Spec2Vec and Random Forest model were trained on the remaining 19 sets. The analogue test sets were generated by splitting the unique 2D structures in the library into 20 equal groups, for which all corresponding spectra were selected. Benchmarking for the exact matches test set was also done by creating 20 test sets. These test sets were generated by selecting all unique 2D structures in the library and randomly selecting one spectrum from these, leaving the rest in the library. The 20 sets differ in which spectrum was randomly selected from the unique 2D structures.

The performance of MS2Query was compared to the performance of the modified cosine score, the cosine score and MS2Deepscore. All four methods use a minimal threshold for the spectral similarity score to determine if a library spectrum is a good analogue or exact match. The threshold for each method was varied between 0 and 1, followed by calculating a performance metric and the recall for predictions falling in the selected threshold. The used performance metric for the "analogue test set" was the Tanimoto score between the predicted library molecule and the correct test molecule. The performance metric for the "exact matches test set" was the percentage of true matches, a prediction was considered a true match when it matches the 2D structure of the correct molecule.

In case of the analogue search using MS2Deepscore and modified cosine score, library spectra were first filtered on a maximum precursor m/z difference of 100 Da. To perform the benchmarking of the search for exact matches using MS2Deepscore and cosine score, only library spectra were considered within a mass tolerance of 0.25 Da. To calculate the cosine score, the cosine greedy implementation of matchms[33] was used with a fragment mass tolerance of 0.05 Da.

### Speed and memory optimization
MS2Query was optimised for speed and working memory efficiency. To make this possible, MS2Query aims to avoid repetitive, computational expensive operations. The biggest speed improvement was achieved by pre-calculating mass spectral embeddings for Spec2Vec and MS2Deepscore. MS2Deepscore and Spec2Vec both predict a chemical similarity score between two library spectra, by first calculating a multidimensional embedding followed by calculating the (mathematical) cosine similarity between these two embeddings. The library spectra are already known, therefore the embeddings for all library spectra are pre-calculated and stored. Therefore, only for the query spectra the embeddings have to be computed, instead of for all the library spectra.

In the first step of MS2Query, the top 2000 library spectra are selected that have the highest MS2Deepscore between the query spectrum and a library spectrum. To do this selection, the MS2Deepscores between a query spectrum and all MS2Deepscores are calculated. To avoid repetitive calculation of these scores, the calculated MS2Deepscores are reused to calculate the average of the MS2Deepscore of multiple chemically similar library molecules.

The precursor m/z is the only metadata entry that is required for MS2Query and which serves to calculate the mass differences. Other spectra metadata such as retention time, SMILES, or compound names can be returned for results found by MS2Query. To reduce the toll on working memory, this information is stored in a SQLite library. The precursor m/z is stored in a separate SQLite library column for efficient look-up speeds. To calculate the average MS2Deepscore of multiple chemically similar library molecules, the 10 most chemically similar library molecules based on the Tanimoto score are needed. This top 10 list of most related InChIKeys is pre-calculated for every unique InChIKey in the library and stored in the SQLite library.

MS2Query contains a workflow to automatically generate all needed files for an MS2Query library and a fully automatic workflow for training all the needed models on a new library. This makes it straightforward to run MS2Query or create models for different mass spectral libraries.

### Speed performance
The speed was tested on the 5987 test spectra in positive mode and compared to the positive mode GNPS library containing 302.514 spectra. The test was run on a laptop; the Lenovo Thinkbook 15-IIL. This laptop has an 11th generation Intel Core i5-1135G7 and 16GB installed RAM.

### Case studies
Four case studies were performed to confirm that MS2Query performs well on newly generated experimental data. Two blood plasma samples, a urine sample and a bacterial sample set were analysed. The raw data, intermediate files and raw results can be found on https://doi.org/10.5281/zenodo.6811540. Here below, the materials, the analytical methods used, and the analytical and data pre-processing and processing steps are described for all case studies.

### Case study 1: NIST Human blood plasma
The NIST 1950 Frozen Human Plasma standard reference material (SRM) was used. The sample was subjected to reversed-phase chromotographic (RPC) assay tailored for complex lipid separation developed by Lewis, et al.[39], as further described below.

### Case study 2: blood plasma long-term reference
For this case study, a plasma Long-Term Reference (LTR) sample was used. This LTR is routinely integrated in profiling studies at the National Phenome Centre for study-independent monitoring of precision. To create the plasma LTR, 10 L of bulk plasma were purchased from Seralab, homogenized, and aliquoted for long-term storage at −80 °C. Hydrophilic interaction liquid chromatography (HILIC) was used in this case study for the analysis of polar metabolites in a sample of plasma LTR[39], as further described below.

### Case study 3: urine long-term reference
For this case study, a pooled Long-Term Reference (LTR) urine sample, maintained by the National Phenome Centre and utilized as an independent sample reference throughout all molecular profiling studies, was used. The protocol followed to generate urine LTR sample is described in detail by Lewis et al.[40]. Briefly, this material was created by

pooling together 78 urine voids collected from healthy volunteers in one day. LTR urine collection was carried out under REC Wales approval: 12/WA/0196. No screening criteria were used to assess the health status of the donors. All samples were combined in a 20 L vessel, homogenized and aliquoted into 15 mL polypropylene conical centrifuge tubes (Corning) for long-term storage at −80 °C. Samples were analyzed by reversed-phase chromatographic (RPC) assay tailored for small molecule metabolites[39], as further described below.

## Pre-processing case study 1

To perform the lipidomic profiling, a combined refence standards mixture (referred to as LipidMix) is added to all samples during protein precipitation with isopropanol (IPA). The composition of the Lipid standard mixture is shown in Supplementary Table 5. The stock solutions of the lipid standards are prepared in and further diluted in the mixture with IPA.

NIST 1950 human plasma sample was thawed at 4 °C for 2 h. Subsequently, a 50 μL aliquot was taken and prepared for lipid analysis by dilution with LC-MS grade water (1:1 v/v) and addition of four parts of isopropanol (IPA) containing a mixture of lipid reference standards[39] to one part of diluted sample for protein precipitation. Vial with the sample was mixed at 1400 rpm for 2 h at 4 °C and subsequently centrifuged for 10 mins at 3486 × $g$ at 4 °C to separate the homogenous supernatant from the precipitated protein. The clear supernatant was aspirated and dispensed into LC-MS vial, then additionally centrifuged for 5 mins at 3486 × $g$ and 4 °C prior analysis. Prepared sample was injected (1 μL) in the chromatographic system using full loop mode (5× overfill).

## Pre-processing case study 2

To perform the HILIC profiling, the method reference (MR) mixture of the reference standards is added to all pooled QC samples providing metabolite targets that represent the wider observable metabolome while facilitating a more real-time assessment of data quality. To allow assessment of sample preparation and injection precision as well as some limited assessment of matrix effect across individual study samples, internal standards (IS) are added to all pooled QC and study samples. When preparing samples for small molecule for HILIC profiling, MR and IS mixtures are prepared in aqueous solutions and added to the sample prior to further preparative steps. The composition of the HILIC MR and IS standard mixture is shown in Supplementary Note 9 (Supplementary Table 6).

Plasma LTR sample was prepared for the analysis by HILIC method in positive ionization mode. The sample was thawed at 4 °C for 2 h. Subsequently, a 50 μL aliquot of plasma LTR sample was diluted 1:1 with LC-MS grade water and a mixture of HILIC MR and IS as described in Supplementary Note 9 (Supplementary Table 6) with 10 μL each. Three parts of acetonitrile were then added to one part of diluted sample for protein precipitation. Vial with the sample was mixed at 1400 rpm for 2 h at 4 °C and subsequently centrifuged for 10 mins at 3486 × $g$ at 4 °C to separate the homogenous supernatant from the precipitated protein. The clear supernatant was aspirated and dispensed into LC-MS vial, then additionally centrifuged for 5 mins at 3486 × $g$ and 4 °C prior analysis. Prepared sample was injected (2 μL) in the chromatographic system using full loop mode (5× overfill).

## Pre-processing case study 3

Preparation and analysis of urine samples are described in detail by Lewis et al.[40]. In brief, an aliquot of 150 μL of urine LTR sample was mixed with RPC-specific MR solution in proportion LTR:MR 2:1 and further diluted with 75 μL of ultrapure water and 75 μL of RPC-specific internal standards (IS) solution shown in Supplementary Note 9 (Supplementary Table 7). The sample was mixed at 850 rpm for one minute at 4 °C and centrifuged for 10 mins at 3486 × $g$ at 4 °C. The supernatant was aspirated and dispensed into LC-MS vials for the

analysis. Urine sample was injected (2 μL) in the chromatographic system using full loop mode (5× overfill).

## UHPLC-MS profiling analysis for case studies 1-3

All UHPLC-MS analyses were performed on Acquity UPLC instruments, coupled to Xevo G2-S TOF mass spectrometers (Waters Corp., Manchester, UK) via a Z-spray electrospray ionization (ESI) source.

To perform the lipid profiling, all solvents – water, acetonitrile (ACN), and IPA and mobile phase additives ammonium acetate and acetic acid were of LC-MS grade. Lipidomic profiling was conducted using a 2.1 × 100 mm BEH C8 column, thermostatted at 55 °C. Mobile phase A consisted of a 2:1:1 mixture of water:ACN:IPA with 5 mm ammonium acetate, 0.05% acetic acid, and 20 μM phosphoric acid. Mobile phase B consisted of 1:1 ACN:IPA with 5 mM ammonium acetate, 0.05% acetic acid. The initial conditions were 99:1 A:B at a flow rate of 0.6 mL/min. The gradient elution program is based on the protocols associated with Lewis et al[39]. and is shown in Supplementary Note 9 (Supplementary Table 8).

The HILIC chromatographic retention and separation of polar molecules was conducted using a 2.1 × 150 mm Acquity BEH HILIC column thermostatted at 40 °C. 20 mM ammonium formate in water with 0.1% formic acid was used as mobile phase A and ACN with 0.1% formic acid as mobile phase B. The initial conditions were 5:95 A:B at a flow rate of 0.6 mL/min. The gradient elution program is based on the protocols associated with Lewis et al[39]. and is shown in Supplementary Note 9 (Supplementary Table 9).

To perform the urine profiling, water and ACN supplemented with 0.1% formic acid of LC-MS grade were used as mobile phases A and B. A 2.1 × 150 mm HSS T3 column thermostatted at 45 °C was used with a mobile phase flow rate of 0.6 mL/min. The gradient elution program is based on the protocols associated with Lewis et al[39]. and is shown in Supplementary Table 10.

The analysis of blood plasma and urine reference samples in presented case studies were performed in positive ionization mode. The mass spectrometry parameters were set as follows: capillary voltage 2 kV for lipid profiling and 1.5 kV for urine profiling, sample cone voltage 25 V for lipid profiling and 20 V for urine profiling, source temperature 120 °C, desolvation temperature 600 °C, desolvation gas flow 1000 L/h, and cone gas flow 150 L/h. Data were collected in centroid mode with a scan range of 50–2000 $m/z$ and 50–1200 $m/z$ for lipid and urine profiling, respectively, and a scan time of 0.1 s. For mass accuracy, LockSpray mass correction was performed using a 600 pg/μL leucine enkephalin solution ($m/z$ 556.2771 in ESI+) in 1:1 water:ACN solution at a flow rate of 15 μL/min. Lockmass scans were collected every 60 s and averaged over 4 scans. The mass spectrometer was operating in Fast DDA mode. The intensity threshold of precursor ion was set to 100 K to trigger MS$^2$ fragmentation that was performed in centroid mode with a scan range of 50–2000 m/z and a scan time of 0.25 s. MS$^2$ was switched back to MS survey function after 2 s acquisition. Deisotoped peak selection option was enabled. The collision energy was set to the ramp of 15–30 eV and 30–60 eV for MS$^2$ acquisition of low and high mass ions, respectively. Ten iterative DDA acquisitions were performed using DDA auto-exclude program, which allows ions selected as precursors in previous injections are removed from the list in the following injections.

## Case study 4: anammox bacteria

For the fourth case study extracts of three strains of anammox bacteria were used. *Kuenenia stuttgartiensis* MBR1 was cultivated in a 12 liter single-cell membrane bioreactor (MBR) as previously described by Kartal et al.[41]. In brief, the growth medium consisted of (per liter): 1 g KHCO3, 0.025 g KH2PO4, 0.6 mM HCl, 45 mM NaNO2, 45 mM (NH4)2SO4, 0.15 g CaCl2·2H2O, 0.1 g MgSO4·7H2O and 0.00625 g FeSO4·7H2O, and 1.25 ml trace elements consisting of (per liter) 0.24 g CoCl2·6

H2O, 0.25 g CuSO4·5 H2O, 0.014 g H3BO3, 0.99 g MnCl2·4 H2O, 0.22 g Na2MoO4·2 H2O, 0.05 g Na2WO4·2 H2O, 0.19 g NiCl2·6 H2O, 0.067 SeO2, 15 g Tritiplex III (EDTA), 0.43 g ZnSO4·7 H2O). In the reactor, the temperature was maintained at 33 °C with a heating jacket and the biomass was stirred at 200 r.p.m. with a six-bladed turbine stirrer. Excess biomass was removed at 1.1 L per day, resulting in a doubling time of 10 days. *Brocadia fulgida* was cultivated as previously descri-bed by Kartal et al[42]. with some adjustments: the working volume was 6 liter and the bacteria were kept in a single-cell membrane bioreactor. The growth medium consisted of (per liter): 1.25 g KHCO3, 0.1 g KH2PO4, 0.048 g MgSO4·7H2O, 0.0576 g CaCl2·2H2O, 0.00625 g FeSO4·7 H2O, 1.25 ml trace elements (as described above), 45 mM NaNO2, and 45 mM (NH4)2SO4. Temperature was regulated with a heating jacket and kept at 33 °C. The biomass was stirred at 200 r.p.m. with a six-bladed turbine stirrer. The reactor was originally inoculated with activated sludge from the secondary stage of the Dokhaven municipal wastewater treatment plant (Rotterdam, The Netherlands). *Scalindua* was cultivated in a 5.5 liter sequencing batch reactor at room temperature as described earlier by van de Vossenberg et al[43]. Medium consisted of (per liter): 30 g red sea salt, 0.003125 g KHCO3, 0.025 mg FeSO4·7 H2O, 30 mM NaNO2, and 30 mM (NH4)2SO4. The biomass was stirred at 350 r.p.m. with a six-bladed turbine stirrer. *Scalindua* was originally enriched from the deepest part of the Gullmar Fjord (Alsbäck, 58°15.5′N, 11°13.5′E, water depth 116 m). Samples (30 mL) were taken from each reactor in triplicate and kept on ice. After cen-trifugation at 3000 × g, at 4 °C for 5 minutes, the cell pellets were lysed in ice-cold acetonitrile:methanol:water (2:2:1; v:v:v). The samples were snap-frozen in liquid nitrogen and stored at −70 °C until further use. To remove precipitated proteins and extracellular matrix, samples were centrifuged again at 20,238 × g, at room temperature for 5 minutes. Subsequently, samples were subjected to LC-MS analysis as described previously by Jansen et al[44]. with several adaptations. The samples were injected onto a Diamond Hydride Type C column and separated using a gradient of acetonitrile and water (both with 0.2% formic acid) on an Agilent 1290 II LC system coupled to an Agilent Accurate Mass 6546 Quadrupole Time of Flight (Q-TOF) instrument operated in the posi-tive ionization mode and a scan range of 50-1200 m/z. For data dependent acquisition of MS2 spectra, automated selection of max-imum 4 precursor ions (>m/z 100) per cycle with an exclusion window of 2 minutes after a single spectrum, and an absolute threshold of 1000 counts with a mass error tolerance of 20 ppm was used. The scan speed was varied based on precursor abundance with a target of 50,000 counts. Common background ions were excluded, the isola-tion width was set to narrow (-1.3 m/z), and the collision energy was set to 20 V. Data collection was performed using Agilent Masshunter software 10.0 (Agilent Technologies).

### Data processing
In the case of case studies 1-3 the spectra were uploaded on GNPS to run MSCluster[45] to create consensus spectra. These consensus spectra were taken as input for MS2Query. The data files for case study 4 were first converted to mzML format using Proteowizard (Chambers et al., 2012). Next, LC-MS features were picked using XCMS3[46] (https://github.com/sneumann/xcms), using the findChromPeaks function. The resulting MS2 spectral MGF file was used to run MS2Query.

Analogues and exact matches with a MS2Query score above 0.633 (corresponding to 35% recall for the "analogues test set" during benchmarking) were considered for all case studies. In addition, an analogue search on the GNPS platform[20] for case studies 1-3 was per-formed and FBMN for case study 4 was performed. More information about this can be found in the Supplementary Note 6.

### Manual validation
To validate the MS2Query matches for case study 1-3, metabolites with MS$^2$ were manually annotated to confidence level 1-3 according to the

Metabolomics Standards Initiative[47] by matching fragmentation spec-tra to reference data from an in-house standards database and online databases LIPID MAPS[48], HMDB[49], and GNPS[20]. In the case of case study 4, annotations were checked based on a combination of biological knowledge and matching of MS1 mass and retention time to reference standards. Judgement of the analogue quality was done manually. Lipids where the lipid type (e.g. PC or SM) was correctly predicted and the chain lengths were similar, were marked as a good analogue. Correctly predicted lipids, but wrong lipid types were marked as analogue. The detailed manual annotations and judgements for all spectra can be found as an excel file in the Supplementary Data 1 for all case studies.

### Reporting summary
Further information on research design is available in the Nature Portfolio Reporting Summary linked to this article.

### Data availability
The models and spectra files used for the case studies can be down-loaded from the Zenodo database at https://doi.org/10.5281/zenodo.6124553. For the k-fold cross-validation the raw results, the raw data and the data splits can be downloaded from the Zenodo database at https://doi.org/10.5281/zenodo.7427094. The mass spectrometry data for the case studies were deposited in the MassIVE repository (https://massive.ucsd.edu/) under accession number MSV000089648 and MSV000090642 [https://doi.org/10.25345/C52B8VH15]. To validate the case study results, multiple libraries were used: LIPID MAPS (https://www.lipidmaps.org/), HMDB (https://hmdb.ca/) and GNPS (https://gnps.ucsd.edu/). Source data are provided with this paper.

### Code availability
MS2Query is available as an easily installable Python library running on Python 3.7 and 3.8 on Windows, Linux and MacOS. Source code and installation instructions can be found on Github (https://github.com/iomega/ms2query). The case study results were obtained using version 0.3.2 and the k-fold cross-validation results were obtained using ver-sion 0.6.6. Version 0.6.6 can also be downloaded from Zenodo at https://doi.org/10.5281/zenodo.7691816[50].

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

## Acknowledgements

The authors would like to thank Dr Marynka Ulaszewska, Dr Lapo Renai, and Huali Xie for sharing experimental data to test the first versions of MS2Query. The authors would also like to thank colleagues from the eScience Center and Prof. Marnix Medema from the WU Bioinformatics Group for useful discussions related to MS2Query. We thank David Joas for providing recent code changes to MS2DeepScore making it possible to include precursor m/z as an additional input feature. The authors are grateful to the Netherlands eScience Center for support through the ASDI eScience grant [grant number ASDI.2017.030] (FH and JJJvdH), to the Medical Research Council UK Consortium for MetAbolic Phenotyping (MAP UK) [grant number MR/S010483/1] (EC and SC), and to the

Infrastructure support provided by the NIHR Imperial Biomedical Research Centre (EC and SC).

## Author contributions
N.F.J., J.J.R.L., F.H., and J.J.J.vdH. initiated the project idea. N.F.J. wrote the main code for MS2Query, did the benchmarking, and wrote the first draft of the paper. F.H. did many code reviews and wrote part of the code of MS2Query. J.J.R.L. built the first prototype for MS2Query. J.J.J.vdH., F.H., and J.J.R.L extensively reviewed and edited the manuscript. E.C. and S.C. shared the data and manually validated the results for case studies 1-3. F.J.V. and R.S.J. collected the data and manually validated the results for case study 4. F.H. and J.J.J.vdH. supervised the work. All authors read and approved the manuscript.

## Competing interests
JJJvdH is currently a member of the Scientific Advisory Board of NAI-CONS Srl., Milano, Italy. All other authors declare no conflict of interest.
