## [Peer Review File · Nature Communications]

REVIEWER COMMENTS

Reviewer #1 (Remarks to the Author):

The paper describes MS2Query, a method for analogue search in spectral libraries.

Given that spectral libraries are and will be vastly incomplete, methods that allow us to derive information about novel compounds (absent from all libraries) are of great interest. The new method MS2Query is, according to the evaluation results of the paper, indeed substantially stronger than all previous approaches. The method is therefore of high significance to the field.

The paper is generally very well written and easy to follow. There are several repetitions of stuff said 2-3 times in subsequent sentences and paragraphs, see below for a few examples; please check the text for further doublets.

The method description is adequate, the evaluations are extensive and meaningful.

Major

* In the evaluation, it is not clear to me if the sets used for training and evaluations are structure-disjoint. (The paper fails to explain what that is, by the way: Even the Online Methods avoids a clear definition and

rather talks about InChI. See below.) Stereoisomers are known to fragment very similarly, etc pp. The analogue test set should, by definition, be structure disjoint: That is, no spectra from any molecular structure (2D) in the analogue set should be contained in the training data. The "dataset workflow" in Fig. 6 does not really help me to decipher what datasets are used for what purpose, and *how they overlap*.

* The important point is: There are three ML methods integrated here (Spec2Vec, MS2DeepScore, MS2Query). For *all three ML models*, training and analogue test set have to be structure-disjoint. If this is the case, then state it somewhere in the main text. If not, how can we trust the evaluation metrics? I could live with Spec2Vec not being trained structure-disjoint, but not with MS2DeepScore.

* Similarly, the exact match test set should be spectrum-disjoint to all three training sets. If this is the case, then state it somewhere in the main text.

* I am not convinced that MS2Query performs substantially better for larger compounds. I believe that to a large extent, this is an artifact of a) the spectral libraries and b) the Tanimoto score. We observe basically the same trends for cosine score and MS2Deep, SFigs 1+2. Hence, I would considerably tone down the discussion of these results, both in the introduction and in the discussion.

Minor

* The paper should clearly state that this is about 2D structures and 2D structural similarity -- at least, that is what I assume. Daylight fingerprints are for 2D structures?

* In that respect, the paper may also explain **why** -- stereoisomers fragmenting so similar etc pp.

* Introduction: "in silico methods" also refers to combinatorial fragmenters (MetFrag, MS-FINDER etc) and methods using molecular fingerprints (CSI:FingerID etc); not just methods that simulate a mass spectrum. Somewhat sloppy, but everybody does it, so...

* p3 l68-69: In particular, all such in silico methods "match against structural libraries" -- that should be "molecular structure databases", I assume

* p3 l86: "manifold" -- "substantially"

* p3 l90-91: citations, Demuth et al, Anal Chim Acta 2004 for EI-MS, Rasche et al, Anal Chem 2012 for MS/MS (somewhere in the supplement) or something earlier if you can find it

* p3 l106-107: "(currently most widely used)" is a repetition

* p4 l145: is matchms know for its speed? If so, mention.

* p5 l157-167: do not use the word "accuracy" unless you mean accuracy

(percentage of correct predictions) -- maybe "quality"?

* p5 Fig2: Threshold is varied but not shown -- good! (no pun, just like ROC)

* p6 Fig3: Showing this figure without explicitly mentioning that SFig 1+2 show exactly the same trend is misleading

* p6 Fig3: I do not like the bar plot here. I suggest to use points instead that, solely for visualization, are connected by lines. So, it will be three (meaningless) lines connecting three points per interval.

* p7 l229: "shows" - "indicates"

* p7 Fig4 subfig 3: I am not happy with the example, a dimer is something special. Furthermore, we could find them without an open search, simply search with masses $2M$ or $1/2M$. To show that the analogue search does something useful even for molecules of very different mass, which this fig is about, a different example should be presented.

* p8 l264: comma after "MS2Query"

* p8 l278: "less well" -- worse!

* p8 l282: "predicts a match" -- does that refer to "an analogue match"?

* p9 l319-323: repetitive, this has been set in the paragraph above

* p10 l349-350: repetitive

* p11 l361: getting the structure from an InChI is not a good idea, structures should be standardized/normalized/whatever you call it. PubChem SMILES do the trick, but rdkit or CDK have functions for that, too, I think.

* p11 l375: mention in main that spectra with few peaks were removed

* p11 l382-389: The description of the datasets is not easy to follow, and Fig6 does not help, either.

* p12 Analogue search: You restricted cosine search to +/- 100 Da, in order to improve its running time, understood -- but does that change anything WRT quality of the results?

* p12 Analogue search: Why did you limit MS2Deep to +/- 100 Da? It does not have any issues with running time! In fact, it is faster than MS2Query by design...

Reviewer #3 (Remarks to the Author):

In this paper de Jonge et al. propose an approach for finding molecular analogues and exact molecular matches for unannotated mass spectra. Building upon models previously published by the group, the authors present a new model, MS2Query. MS2Query works by first precomputing embeddings for library spectra using the groups' prior models MS2Deepscore and Spec2Vec. For a given query spectrum, library spectra are filtered based on MS2Deepscore embedding similarity. A

handful of features are then calculated for the query spectrum and remaining library spectra and fed into a random forest model trained to predict Tanimoto scores between the molecules underlying each query-spectrum-library-spectrum pair. The library spectrum with the highest such score is selected as the best match.

The method is validated on two validation sets, one in which each query spectrum has at least one spectrum in the library from the exact same molecule and one with no such matches. The authors show that on both test sets, MS2Query selects more similar analogues and a higher percentage of exact matches than MS2DeepScore or modified cosine score alone. They also show that by using precomputed embeddings, MS2Query greatly outperforms modified cosine score in terms of throughput. The model is also validated on 4 use cases.

The paper has several issues that need to be addressed:

1. The paper does not provide a compelling argument why we need a new method such as MS2Query is needed instead of improving the performance of MS2DeepScore to increase the scores for analogues. Anyone versed in Deep Learning recognizes that it is very much suited to perform both exact matches and analogue matches.
2. The limitations of MS2DeepScore that this paper addresses are not quantified. Can you re-run MS2DeepScore with added features such as query precursor m/z, precursor m/z, and Spec2Vec Similarity and report the results in this paper to justify the need for MS2DeepScore?
3. The value of this work is not well articulated. Please add a paragraph to explain this clearly earlier in the document. It's not until the caption in Figure 4 that we learn that, "this makes it possible to also detect analogues that have large mass differences (such as dimers for instance)." But then this raises the question if MS2DeepScore can be trained on analogues with large mass differences.
4. Reference to prior works: "by using fragmentation trees¹⁰⁻¹², or by using in silico methods to match against structural libraries¹³⁻¹⁵." The fragmentation trees are also in silico, so the distinction/categorization of these methods is not clear.
5. Reference to prior works: The paper does not discuss the latest publications that use deep learning to do annotation (MassFormer from Toronto, ESP from Tufts, MassGenie from Liverpool).
6. Reference to prior works: "However, a limitation of the cosine score (and its derivatives) is that small chemical modifications can, and multiple chemical modifications will, often result in a large decrease in mass spectral similarity which limits its ability to serve as a proxy for chemical similarity". The prior work on spectral alignment (reference 16) is not sufficiently discussed and how it addresses this issue.
7. Figure 2 and related section: "Optimal" is not discussed or defined. "Random" is not defined. The (large) gap between optimal and the other methods is not explained. When "recall" is high,

MS2Query's outperformance wanes (or is non-existent?) relative to existing methods. An interpretation is warranted. Why is the first 14 characters of the InChiKeys sufficient for a match?

8. Figure 3 (b), a minimal threshold of 0.633 is arbitrary (and so is a 35% recall). Is it possible to plot these results as you did in Figure 2 to better observe the tradeoffs between recall and accuracy across the different methods. The supplement figure states that this trend is the same for all method), so it is not clear why in the main document only MS2Query is highlighted. The explanation why these methods perform better is not satisfactory. In this section and is only clarified in the Discussion. Dividing the results across Figure 3, S1 and S2 does not allow for a close comparison across all three methods.

9. In the discussion, the authors write: "Reference to prior works: The paper does not discuss the latest publications that use deep learning to do annotation". This statement must be qualified by (identical) performance of all three tools at high recall.

10. How are features 4 and 5 correlated? Is it possible to have a plot showing the correlation and a discussion in this regard?

11. Figure 6 and S6. Can you address the data leakage issues between ms2deepscore and spec2vec and MS2Query?

12. The authors pre-filter spectra based on precursor mass difference for the modified cosine score model, when modified cosine score should already take this sort of metadata into account. It is not made clear why this pre-filtering is necessary.

13. The authors do not discuss the "difficulty" in the test dataset. 5-fold validation and reporting average performance is warranted. Also, doing splits where we see more challenging data splits would be of interest.

14. The modified cosine score is not defined. A section in the supplementary that gives a mathematical formulation would be helpful.

15. P 5, line 159: "accuracy" is not defined

Supplementary material

16. "The model using the feature that first calculates the average MS2Deepscore for each selected library structure, followed by taking the average of the 10 average scores for each library structure performed the best." So you are using many more spectra for the method that you selected? So averaging many more items is better? Please clarify this issue and potentially remove this section as

it seems trivial. It would have been more interesting to see the impact of selecting 5 or 20 items instead of just 10.

17. The results for the negative ionization mode are surprising, especially for the exact match. Can you validate the results on the positive mode by training on a similar dataset size and see if the performance drops as it did for negative mode? Can you also pull this result into the main paper ?

Minor issues

18. When you used this term, “(Modified) cosine scores”, what is the intention? Are you referring to both cosine and modified cosine scores? Please clarify.

19. “the computation time to determine the fragmentation trees also increases manifold.” This sentence can be made more precise regarding the runtime for fragmentation trees

20. “The percentage of true positives is given, a match is marked as true positive if the first 14 characters of the InChiKeys are identical.” Incorrect punctuation. This is not a good English sentence.

21. “These methods perform especially well at predicting chemical similarity for molecules that are similar, but are chemically not exact matches. This makes these scores very suitable for an analogue search.” The last sentence is not clear. Which scores?

22. “your query spectra” – may just called it “the query spectra”? this appears in multiple places in the text.

23. “For all other benchmarking only the positive mode spectra are used”. What other benchmarking is being referred to? Please rewrite to clarify.

We thank both anonymous reviewers and the editor for their positive words on our contribution as well as the constructive remarks on our manuscript. We think both MS2Query as a tool and in particular the manuscript became better as a result. Hence, we very much appreciate your invested time in shaping MS2Query. Below here, we will provide a point-by-point response.

We look forward to hearing back on what you think of the MS2Query tool and manuscript now,

With kind regards, on behalf of all authors,

Niek de Jonge, Florian Huber & Justin van der Hooft

Reviewer #1 (Remarks to the Author)

The paper describes MS2Query, a method for analogue search in spectral libraries.

Given that spectral libraries are and will be vastly incomplete, methods that allow us to derive information about novel compounds (absent from all libraries) are of great interest. The new method MS2Query is, according to the evaluation results of the paper, indeed substantially stronger than all previous approaches. The method is therefore of high significance to the field.

The paper is generally very well written and easy to follow. There are several repetitions of stuff said 2-3 times in subsequent sentences and paragraphs, see below for a few examples; please check the text for further doublets.

The method description is adequate, the evaluations are extensive and meaningful.

We thank the reviewer for the positive words on our work, and the constructive comments below that allowed us to further clarify our approach and prompted us to do several additional analyses that underpin our previous results. Please find our point-by-point reply below.

Major

1. In the evaluation, it is not clear to me if the sets used for training and evaluations are structure-disjoint. (The paper fails to explain what that is, by the way: Even the Online Methods avoids a clear definition and rather talks about InChI. See below.) Stereoisomers are known to fragment very similarly, etc pp. The analogue test set should, by definition, be structure disjoint: That is, no spectra from any molecular structure (2D) in the analogue set should be contained in the training data. The "dataset workflow" in Fig. 6 does not really help me to decipher what datasets are used for what purpose, and *how they overlap*.

Indeed, we can confirm that the training and evaluation set for the analogue test are structure-disjoint. We do acknowledge that this was not clear enough in the previous manuscript version. Thus, we have now highlighted this clearer both in the main text and in the material and methods (see tracked changes manuscript for exact sentences and places).

We also recognized that the old workflow in Fig. 6 was a bit confusing (we refer to point 26 for an updated version of Fig. 6). We rewrote the Material and Methods section to make this part of the workflow clearer. Please note that the workflow for benchmarking has also been changed, since it was replaced by 20-fold cross validation as suggested by reviewer #3 in comment 13 (we refer to there for related changes made in our revision).

2. The important point is: There are three ML methods integrated here (Spec2Vec, MS2DeepScore, MS2Query). For *all three ML models*, training and analogue test set have to be structure-disjoint. If this is the case, then state it somewhere in the main text. If not, how can we trust the evaluation metrics? I could live with Spec2Vec not being trained structure-disjoint, but not with MS2DeepScore.

We can confirm that the test set for the analogue test set was and still is structure-disjoint with the training set of MS2Deepscore and MS2Query (see also our reply to the previous point). All the annotated spectra used for training Spec2Vec were also structurally disjoint. However, we do note that Spec2Vec was potentially not trained fully structure-disjoint: since Spec2Vec is an unsupervised method. To fully make use of the unsupervised nature of Spec2Vec, we used as many mass spectra in the GNPS library as possible, including those that were not annotated with a SMILES or Inchi. These unannotated spectra could therefore potentially correspond to the same structure as spectra in the test set (although we only expect a small overlap here, thus we do not regard this to be an issue). These points are now stated more clearly both in the results section as well as in the material and methods section (see tracked changes manuscript for exact sentences and places).

3. Similarly, the exact match test set should be spectrum-disjoint to all three training sets. If this is the case, then state it somewhere in the main text.

The exact match test set was spectrum-disjoint to all three training sets. We have now clarified this in the Results section and Online Methods (see tracked changes manuscript for exact sentences and places).

4. I am not convinced that MS2Query performs substantially better for larger compounds. I believe that to a large extent, this is an artifact of a) the spectral libraries and b) the Tanimoto score. We observe basically the same trends for cosine score and MS2Deep, S Figs 1+2. Hence, I would considerably tone down the discussion of these results, both in the introduction and in the discussion.

We notice that we did not convey the message that we hoped to convey with the statements about the performance for large(r) masses. The point we hoped to make was that analogue search methods in general seem to perform better for large(r) masses. We did not have the intention to use this as an argument as to why MS2Query is better than MS2Deepscore or using the modified cosine score. Instead, we hoped to highlight the general strength of library search methods for larger masses, since this is complementary with fragmentation tree-based methods that are reported to be especially strong for the small(er) metabolites.

Prompted by this remark, we did several additional tests to see if and quantify how much this “apparent performance gain” is an artifact of the spectral library and the Tanimoto score. In Figure

S3, these results are shown (also displayed below here). The average Tanimoto score for randomly selected spectra from the library show a higher Tanimoto score for test spectra with a higher mass as well. These results indeed support the reviewer's statement that this difference is to some extent an artifact of the Tanimoto score and the spectral library used. We do note that the difference is still larger in the analogue search methods than in the random search.

Following the reviewer's remark, and the outcome of our additional analyses, we moved these results to the supplementary information and removed the statements on this from the main text (introduction, results, and discussion sections). In the previous version of the manuscript, we used bar plots to visualize these results, in the new Figure S3 we changed this to a quality versus recall plot, as requested in comment 8 by reviewer 3 and in addition used the 20-fold cross-validation test sets. We think our analyses and observations have value for readers that want to get more in-depth insight in the performance of analogue searches and therefore think this fits well in the supplementary information, but is not relevant enough for the main text.

Figure S3: Recall vs quality plots, with test sets split on mass bins. These are the same test sets used for the "analogue test set" in the k-fold cross validation, but here each test set is split on the precursor m/z of the test spectrum.

Minor

5. The paper should clearly state that this is about 2D structures and 2D structural similarity -- at least, that is what I assume. Daylight fingerprints are for 2D structures?

Indeed, 2D structures were used, this is clarified in both the main text and Online Methods.

6. In that respect, the paper may also explain **why -- stereoisomers fragmenting so similar etc pp.**

We added an explanation at the start of the results section: "2D structures were used, since tandem mass spectrometry cannot discriminate between different stereoisomers, since they yield similar or identical fragmentation mass spectra."

7. Introduction: "in silico methods" also refers to combinatorial fragmenters (MetFrag, MS-FINDER etc) and methods using molecular fingerprints (CSI:FingerID etc); not just methods that simulate a mass spectrum. Somewhat sloppy, but everybody does it, so...

and

8. p3 l68-69: In particular, all such in silico methods "match against structural libraries" -- that should be "molecular structure databases", I assume

We do apologize for any confusion here and rephrased the sentence into: "Currently, three main types of approaches to determine molecular structures from MS² spectra exist: matching against annotated mass spectral library spectra⁴⁻⁹, by using fragmentation trees¹⁰⁻¹², or by predicting fragmentation mass spectra from chemical structures to match against molecular structure databases¹³⁻¹⁷. However, all these approaches still have important limitations."

9. p3 l86: "manifold" -- "substantially"

Rephrased.

10. p3 l90-91: citations, Demuth et al, Anal Chim Acta 2004 for EI-MS, Rasche et al, Anal Chem 2012 for MS/MS (somewhere in the supplement) or something earlier if you can find it

We added the citations to line 90-91.

11. p3 l106-107: "(currently most widely used)" is a repetition

We removed "(currently most widely used)".

12. p4 l145: is matchms know for its speed? If so, mention.

Yes, indeed, the matchms implementation to compute spectral similarities is optimized for performance. This is now also mentioned in the results section.

13. p5 l157-167: do not use the word "accuracy" unless you mean accuracy (percentage of correct predictions) -- maybe "quality"?

We have rephrased accuracy into quality throughout the entire manuscript, including the figures.

14. p5 Fig2: Threshold is varied but not shown -- good! (no pun, just like ROC)

Thanks, that is good to hear!

15. p6 Fig3: Showing this figure without explicitly mentioning that SFig 1+2 show exactly the same trend is misleading

In an attempt to make the presentation less ambiguous, we moved Figure 3 to the supplementary information, where it can be directly compared to the performance of MS2Deepscore and Modified cosine score. We refer (back) to the 4th comment for more details.

16. p6 Fig3: I do not like the bar plot here. I suggest to use points instead that, solely for visualization, are connected by lines. So, it will be three (meaningless) lines connecting three points per interval.

We agree that this bar plot can be confusing. As a solution, we changed this to a quality versus recall plot (like in figure 2), as requested in comment 8 by reviewer 3. This also gives more consistency throughout the paper, (hopefully) making it easier for the reader to follow. We refer back to our answer to point 4 for more details and the new version of this figure.

17. p7 l229: "shows" - "indicates"

Rephrased.

18. p7 Fig4 subfig 3: I am not happy with the example, a dimer is something special. Furthermore, we could find them without an open search, simply search with masses 2M or 1/2M. To show that the analogue search does something useful even for molecules of very different mass, which this fig is about, a different example should be presented.

We agree that a dimer is a special case and that this was perhaps not the best example to show the average use case of MS2Query. We have now selected a few different examples that showcase the type of analogues MS2Query found in the case studies better, that include a positional isomer and

two analogues with a smaller (<50 Da) and larger (>160 Da) m/z difference (see changed part of the Figure 3 below).

Examples case studies				
3	Manual annotation	Analogue predicted by MS2Query	m/z difference	MS2Query prediction
	 Paraxanthine	 Aminophylline	0,0 Da	0,88
	 PEG(n8)	 Tetraglyme	165,1 Da	0,79
	 LPC(20:4/0:0)	 LPC(16:0/0:0)	48,0 Da	0,93

19. p8 l264: comma after "MS2Query"

Rephrased.

20. p8 l278: "less well" -- worse!

Rephrased.

21. p8 l282: "predicts a match" -- does that refer to "an analogue match"?

A "match" here refers to both, since the prediction can be both an analogue or an exact match (depending on the mass difference between query and hit). We agree that a match suggests that it is a good prediction, but that is not what we meant here, we were just referring to the recall.

Therefore, the sentence was changed into: "In the four case studies the recall varies from 15% to 75% percent with the same settings (Figure 4)."

22. p9 l319-323: repetitive, this has been set in the paragraph above

We removed the entire paragraph.

23. p10 l349-350: repetitive

We removed the following sentence: "The MS2Deepscore between these library spectra and the query spectrum is calculated and the average per library structure is taken."

24. p11 l361: getting the structure from an InChI is not a good idea, structures should be standardized/normalized/whatever you call it. PubChem SMILES do the trick, but rdkit or CDK have functions for that, too, I think.

Thanks for noticing this. Actually, this was a small mistake in the Materials and Methods section, since we did not use InChIs for the fingerprint selection, but we used SMILES. The SMILES used were first “sanitized” by RDKit before generating the molecular fingerprint. This has now been corrected in the Materials and Methods.

25. p11 I375: mention in main that spectra with few peaks were removed

Following this suggestion, we added the following sentence to the Results section under “MS2Query has a high accuracy for analogue searching and exact matching in benchmarking”:

“These spectra were first cleaned and filtered, including cleaning metadata, filtering out spectra with less than 3 peaks and normalizing intensities.”

26. p11 I382-389: The description of the datasets is not easy to follow, and Fig6 does not help, either.

We recognize that the old workflow in Figure 6 was confusing. We rewrote the Material and Methods section to make the workflow description (hopefully) clearer to the reader. The workflow for benchmarking has also been changed, since it was replaced by 20-fold cross validation as suggested by reviewer #3 in comment 13. We added the new version of Figure 6 below.

Figure 6: Workflow for training the MS2Deepscore model, the Spec2Vec model and the random forest model used by MS2Query.

27. p12 Analogue search: You restricted cosine search to +/- 100 Da, in order to improve its running time, understood -- but does that change anything WRT quality of the results?

This setting was chosen, since this is part of the default settings in tools like Masst and GNPS for an analogue search. To visualize the effect this has on the quality of the results, we did run the cosine score without any preselection on parent mass on one of the test sets of k fold cross validation. The results can be found in Supplementary Figure S8. The performance difference for all three methods is only small, when including or excluding a preselection on 100Da. For the cosine score and MS2Deepscore the performance actually slightly drops when no preselection on 100 Da is performed. For the modified cosine score the performance improves a bit for settings that result in a low recall. These results fit our expectation, as some typical analogs such as glucosides or glucuronides might be found when using this wider mass range; however, since more comparisons must be made that can also result in more false positives. In addition, the runtime increases quite a bit. It took over 70 hours to run this for one of the k-fold cross validation test sets (17135 spectra) for the cosine score and even longer for the modified cosine score.

These results were added in the supplementary information section S8:

“S8. Performance without precursor mz preselection

Benchmarking with Cosine score, Modified cosine score and MS2Deepscore was done with a preselection on mass difference between a query and library molecule of 100 Da. In Figure S8, we compare the difference in performance with and without a preselection on mass difference. These results show that performance is slightly better when using prefiltering on a mass difference of 100 Da, when using MS2Deepscore and the cosine score. When using the modified cosine score the results are a bit better for low recall settings, when no preselection on mass difference is done, however this comes at the cost of a large increase in runtime.”

Figure S8: Comparisons of performance for methods with and without preselection on 100Da. a: Modified Cosine score with a preselection on a mass difference of 100 Da performs slightly better on low recall. The first test set of the 20-fold cross validation was used. b: Cosine score with a preselection on a mass difference of 100 Da performs slightly better than the cosine score without any preselection on mass difference. The first test set of the 20-fold cross validation was used. c: MS2Deepscore with preselection on a mass difference of 100 Da performs slightly better than MS2Deepscore without any preselection on mass difference. The same 20-fold cross validation test sets are used as in the main text.

28. p12 Analogue search: Why did you limit MS2Deep to +/- 100 Da? It does not have any issues with running time! In fact, it is faster than MS2Query by design...

Actually, this setting was not implemented for running time improvements, instead it was done for performance improvement. Doing a preselection on 100 Da improved the performance of MS2Deepscore. Supplementary Figure S8 was added to support this. See the point above (#27) for more details.

Reviewer #3 (Remarks to the Author):

In this paper de Jonge et al. propose an approach for finding molecular analogues and exact molecular matches for unannotated mass spectra. Building upon models previously published by the group, the authors present a new model, MS2Query. MS2Query works by first precomputing embeddings for library spectra using the groups' prior models MS2Deepscore and Spec2Vec. For a given query spectrum, library spectra are filtered based on MS2Deepscore embedding similarity. A handful of features are then calculated for the query spectrum and remaining library spectra and fed into a random forest model trained to predict Tanimoto scores between the molecules underlying each query-spectrum-library-spectrum pair. The library spectrum with the highest such score is selected as the best match.

The method is validated on two validation sets, one in which each query spectrum has at least one spectrum in the library from the exact same molecule and one with no such matches. The authors show that on both test sets, MS2Query selects more similar analogues and a higher percentage of exact matches than MS2Deepscore or modified cosine score alone. They also show that by using precomputed embeddings, MS2Query greatly outperforms modified cosine score in terms of throughput. The model is also validated on 4 use cases.

We thank the reviewer for the accurate summary of our work and the constructive comments that helped us to clarify several aspects of our work, and prompted us to do additional analyses, for example to demonstrate the generalizability of our approach. Please find our point-by-point answer below.

The paper has several issues that need to be addressed:

1. The paper does not provide a compelling argument why we need a new method such as MS2Query is needed instead of improving the performance of MS2DeepScore to increase the scores for analogues. Anyone versed in Deep Learning recognizes that it is very much suited to perform both exact matches and analogue matches.

While a method like MS2Deepscore would indeed be suitable for adding precursor m/z, it is by design not possible to add the average MS2Deepscore of multiple library structures into the model. While features related to precursor m/z and Spec2Vec also improve the performance to some extent, the main boost in performance is due to the feature that uses the average MS2Deepscore of multiple library spectra, which we show in Supplementary Tables S1 and S2.

To clarify and better highlight the observations described above, the following section was added in the introduction:

“Current implementations of an analogue search only consider one library spectrum to predict chemical similarity. However, other chemically closely related library structures are expected to have similar structures to a query spectrum as well for a good analogue. MS2Query uses this principle to improve prediction quality of an analogue search, by also using MS2Deepscore for similar library

structures to predict if a molecule is a good analogue. In addition, MS2Query combines the strength of both MS2Deepscore and Spec2Vec, and uses precursor m/z to further improve prediction quality.”

In the Results section we added:

“The feature that has the largest impact on the increased performance is the *Average MS2Deepscore of multiple library molecules*, see Supplementary S2. This feature builds on the following principle: for two library molecules that are chemically very similar, it is expected that if one of these library molecules is a good analogue to the query molecule, the other is a good analogue as well. Hence, it is expected that the MS2Deepscore between the spectrum of a chemically similar library molecule and the query spectrum is also high, for a good analogue that is. MS2Query is the first mass spectral library searching method that uses this principle to re-rank candidate molecules.”

In addition, we now also discuss the results in Supplementary Tables S1 and S2 in the discussion:

“The most important feature for the performance of MS2Query is using the average MS2Deepscore of multiple library structures, this shows the value of using multiple library spectra for predicting good analogues.”

It might be worth noting that we initially used a small deep learning model instead of a Random Forest model for MS2Query. However, we decided to switch to a random Forest model, since the performance was similar and a random Forest gives more insight into the feature importance compared to a deep learning model.

We thank the reviewer for this comment as it helped us to further clarify our approach in the manuscript.

2. The limitations of MS2DeepScore that this paper addresses are not quantified. Can you re-run MS2DeepScore with added features such as query precursor m/z, precursor m/z, and Spec2Vec Similarity and report the results in this paper to justify the need for MS2DeepScore?

We thank the reviewer for this suggestion. Indeed, it is likely that the performance of MS2Deepscore will improve by adding precursor m/z as an input. Adding Spec2Vec or the precursor m/z difference to an MS2Deepscore model is not possible by the design of MS2Deepscore, since these scores are calculated between two spectra. MS2Deepscore uses a Siamese Neural network, which is designed to first calculate embeddings, this embedding calculation process is separated from the actual comparison between two spectra/embeddings; therefore, any score calculated between two spectra is not possible as an input (like Spec2Vec and precursor m/z difference). Nevertheless, we did train a MS2Deepscore model including the precursor m/z as input during training.

This shows that the performance of MS2Deepscore slightly increases if the precursor m/z is also used as input to the neural network. However, the difference is only small and the use of MS2Query results in a substantial further improvement, especially in the area of most importance for practical use, which is around 40 - 10 % Recall. This fits well with our expectations, since the results in Supplementary

Tables S1 and S2 showed that the precursor m/z was used by the model, but that it was not the most important feature. The most important feature is the average MS2Deepscore of multiple library structures.

Analogues test set

Rebuttal Figure (not in main text) Test results of the 1st test set of the 20 fold cross validation test set. MS2Deepscore is the standard MS2Deepscore model used in the benchmarking. MS2Deepscore trained with m/z is the performance of a newly developed MS2Deepscore model using the MZ as input as well.

3. The value of this work is not well articulated. Please add a paragraph to explain this clearly earlier in the document. It's not until the caption in Figure 4 that we learn that, "this makes it possible to also detect analogues that have large mass differences (such as dimers for instance)." But then this raises the question if MS2DeepScore can be trained on analogues with large mass differences.

We think that the main value of MS2Query over MS2Deepscore is that it predicts analogues that are chemically more similar, and the fully automatic workflow for library searching resulting in both exact matches and analogs. Indeed, MS2Deepscore in its current form is already able to predict analogues with large mass differences, by running MS2Deepscore without any preselection on precursor m/z (see also Supplementary Figure S8). The improved performance of MS2Query is to some extent due to adding precursor m/z difference as input to the random forest, but using the average MS2Deepscore of multiple library structures has the biggest impact on the performance of MS2Query. Following this remark and others, we have added more emphasis on the importance of the average MS2Deepscore score over multiple library structures, we hope that the reason for the improvements of MS2Query becomes clearer. Additional explanation was added to the Introduction and Discussion sections (see added parts under point 1).

4. Reference to prior works: “by using fragmentation trees 10-12, or by using in silico methods to match against structural libraries 13-15.” The fragmentation trees are also in silico, so the distinction/categorization of these methods is not clear.

We changed this sentence following suggestions from both reviewers into:

“Currently, three main types of approaches to determine molecular structures from MS² spectra exist: matching against annotated mass spectral library spectra⁴⁻⁹, by using fragmentation trees¹⁰⁻¹², or by predicting mass fragmentation spectra from chemical structures to match against molecular structure databases¹³⁻¹⁷. However, all these approaches still have important limitations.”

5. Reference to prior works: The paper does not discuss the latest publications that use deep learning to do annotation (MassFormer from Toronto, ESP from Tufts, MassGenie from Liverpool).

We added the 3 references to “or by predicting mass fragmentation spectra from chemical structures to match against molecular structure databases¹³⁻¹⁸.”

We have now also referred to our recent review where we clarify many of these approaches, in particular those using machine learning and deep learning. Many of these methods were recently reviewed by our group, in particular those using machine learning¹⁹.”

6. Reference to prior works: “However, a limitation of the cosine score (and its derivatives) is that small chemical modifications can, and multiple chemical modifications will, often result in a large decrease in mass spectral similarity which limits its ability to serve as a proxy for chemical similarity”. The prior work on spectral alignment (reference 16) is not sufficiently discussed and how it addresses this issue.

To further articulate the previous work mentioned by the reviewer, we changed the related paragraph into:

“A different approach to increase the percentage of spectra for which chemical information can be retrieved is by searching for analogues instead of exact matches^{4, 9, 25-27}. This approach also relies on annotated mass spectral libraries, but aims at finding molecules that are chemically similar, without the need for them to be identical. For an analogue search, it is important to have a spectral similarity score that serves as a good proxy for chemical similarity even if two molecules are not identical. A first improvement made in this direction was the development of the modified cosine score, which in contrast to the cosine score also uses neutral losses for determining spectral similarity. This makes the modified cosine score less sensitive to a small chemical modification. However, multiple small chemical modifications can still result in a large decrease in mass spectral similarity, which limits its ability to serve as a proxy for chemical similarity. Recently, two machine learning-based methods were developed that outperform the modified cosine score in predicting chemical similarities from MS² mass spectral pairs; the unsupervised Spec2Vec²⁹ and the supervised MS2DeepScore³⁰. We hypothesised that their chemical similarity predictions offer great potential for performing a reliable analogue search.”

7. a Figure 2 and related section: “Optimal” is not discussed or defined. “Random” is not defined.

Thanks. We have now added to the caption of Figure 2: “The random results show the results if random matches would be selected, and the optimal results show the performance if the best structural match in the library was selected.”

7b. The (large) gap between optimal and the other methods is not explained.

There is indeed still quite some room for improvement between optimal and the performance of MS2Query.

In the discussion we discuss this gap between optimal and MS2Query: “Even though we demonstrate that MS2Query is able to rapidly provide reliable analogues for unknown substances, there is still room for improvement. The current version was trained using available data from GNPS. While a very valuable resource, we do expect that our models will notably improve when our library is built from larger and chemically more diverse datasets.”

7c. When “recall” is high, MS2Query’s outperformance wanes (or is non-existent?) relative to existing methods. An interpretation is warranted.

We believe the most relevant region for any user will be between 40% and 10% recall. The very high recall means that there is no filtering on the predicted score for the predicted analogue, so a cosine score or MS2Query score of 0.3 will still be considered as an analogue in these cases. This is not a threshold we expect users to ever use.

The fact that MS2Query does better for a higher threshold suggests that MS2Query does better than other methods by filtering out unreliable matches.

7d. Why is the first 14 characters of the InChiKeys sufficient for a match?

The first 14 characters of an InChiKeys determine the 2D structure of a molecule, the additional characters are used to specify the 3D structure of a molecule. We only focus on 2D structure, since it is often impossible to differentiate between stereoisomers based on MS/MS spectra alone. This was clarified clearer in the text see the first two minor comments of reviewer 1 as well.

8. Figure 3 (b), a minimal threshold of 0.633 is arbitrary (and so is a 35% recall). Is it possible to plot these results as you did in Figure 2 to better observe the tradeoffs between recall and accuracy across the different methods. The supplement figure states that this trend is the same for all method), so it is not clear why in the main document only MS2Query is highlighted. The explanation why these methods perform better is not satisfactory. In this section and is only clarified in the Discussion. Dividing the results across Figure 3, S1 and S2 does not allow for a close comparison across all three methods.

Following this point, Figure 3 was recreated in the style of Figure 2 and moved to the supplementary information. Thereby having all results next to each other. We decided to move the Figure to the

Supplementary Materials, since this is not one of our main findings, therefore this fits better in the supplementary information.

9. In the discussion, the authors write: “Reference to prior works: The paper does not discuss the latest publications that use deep learning to do annotation”. This statement must be qualified by (identical) performance of all three tools at high recall.

We could not trace back the sentence/statement the reviewer is referring to. Nevertheless, we would like to make it clear that we did performance benchmarking with the widely-used modified cosine score (i.e., the current “gold standard”), and MS2DeepScore alone to showcase improvements over the individual MS2DeepScore results.

10. How are features 4 and 5 correlated? Is it possible to have a plot showing the correlation and a discussion in this regard?

The rationale to add feature 4 was that there is a chance that there are no closely related matches in the library (low Tanimoto score). In these cases it is also unlikely that the average MS2DeepScore over these “similar” library molecules is high, even if it is a good analogue. The reason for adding feature 4 is that the random forest might be able to learn that a less high score is needed for such cases.

Below a plot can be found showing the correlation between feature 4 and 5. There is clearly a correlation between the average ms2deepscore and the average Tanimoto score. We expect that the good analogues are close to the right diagonal in this plot. The combination of feature 4 and 5 makes it possible for the random forest model to learn this.

To get a bit more insight in the distribution of feature 4, we also created a histogram showing the distribution of the average Tanimoto score, this shows that for the majority of spectra the average Tanimoto score is high, but there are certainly cases with low average Tanimoto scores.

In addition, it is good to realize that the impact of feature 4 on the model performance is minimal, as can be seen in supplementary Table S1 and S2.

11. Figure 6 and S6. Can you address the data leakage issues between ms2deepscore and spec2vec and MS2Query?

Test spectra were not used for any training of MS2Deepscore and Spec2Vec. The training spectra were structure-disjoint with the analogue test set and spectrum-disjoint with the exact match test set. This was the case for the training data of the three models, therefore there is no risk of data leakage. We recognize that this was not clarified well in the text. We added clarification both in the main text and rewrote the part about data splitting in the material and methods. See also the first 3 comments of reviewer 1, since these comments are closely related.

12. The authors pre-filter spectra based on precursor mass difference for the modified cosine score model, when modified cosine score should already take this sort of metadata into account. It is not made clear why this pre-filtering is necessary.

The modified cosine score takes in the precursor m/z for determining losses; however, the modified cosine score does not take precursor mass difference between the two spectra into account. It is common practice to do prefiltering on precursor m/z difference to increase performance and to decrease the computational time needed. Prefiltering on 100 Da for modified cosine score is common practice on platforms like MASST and GNPS.

To visualize the effect this has on the quality of the results, we did run the 3 benchmarking methods without any preselection on parent mass on one of the test sets of k fold cross validation. The results can be found in Supplementary Figure S8. The performance difference for all three methods is only small, when including or excluding a preselection on 100 Da. For the cosine score and MS2Deepscore the performance actually slightly drops when no preselection on 100 Da is performed. For the modified cosine score the performance improves a bit for settings that results in a low recall. These results fit our expectation, some typical analogs such as glucosides or glucuronides might be found when using this wider mass range, however since more comparisons have to be made that can also result in more false positives. In addition, the runtime increases quite a bit. It took over 70 hours to run this for one of the k-fold cross validation test sets (17135 spectra) for the cosine score and even longer for the modified cosine score.

In addition we did run an analysis for MS2Deepscore and Cosine score without any preselection on precursor m/z difference and showed that the performance of both methods slightly drops without this preselection, justifying the use of this prefiltering step. See also comment 27 of reviewer 1.

These results were added in the supplementary information section S8:

“S8. Performance without precursor m/z preselection

Benchmarking with Cosine score, Modified cosine score and MS2Deepscore was done with a preselection on mass difference between a query and library molecule of 100 Da. In Figure S8, we compare the difference in performance with and without a preselection on mass difference. These results show that performance is slightly better when using prefiltering on a mass difference of 100 Da, when using MS2Deepscore and the cosine score. When using the modified cosine score the

results are a bit better for low recall settings, when no preselection on mass difference is done, however this comes at the cost of a large increase in runtime.”

Figure S8: Comparisons of performance for methods with and without preselection on 100Da. a: Modified Cosine score with a preselection on a mass difference of 100 Da performs slightly better on low recall. The first test set of the 20-fold cross validation was used. b: Cosine score with a preselection on a mass difference of 100 Da performs slightly better than the cosine score without any preselection on mass difference. The first test set of the 20-fold cross validation was used. c: MS2Deepscore with preselection on a mass difference of 100 Da performs slightly better than MS2Deepscore without any preselection on mass difference. The same 20-fold cross validation test sets are used as in the main text.

13. a The authors do not discuss the “difficulty” in the test dataset. 5-fold validation and reporting average performance is warranted.

We agree that doing cross-validation is indeed a valuable way to show how well the model generalizes. To demonstrate this, we replaced the original benchmarking with benchmarking done using 20-fold cross-validation. To make this possible and aid in future re-trainings of MS2Query, we create a fully automatic pipeline for training all models and creating the mass spectral libraries. Previously, this was done in separate notebooks. We also released this pipeline for fully automatic new model training, which will make it easier for users to train models on other (in house) mass spectral libraries.

We used the 20 different test sets to add the standard deviation to the Figures. We can demonstrate that the results are quite similar to the previously observed benchmarking making MS2Query a generalizable approach.

We thank the reviewer for this suggestion and think this really is an important improvement, since it shows the generalizability of the approach of MS2Query.

We added the largest changes to the manuscript below, but also made other smaller changes in the manuscript to adjust for the change to 20-fold cross validation.

The results are replaced with the results of the 20-fold cross validation. The results of the old model have been moved to Supplementary Information S1.

Figure 2: MS2Query is more accurate for finding analogues than using MS2Deepscore or modified cosine score and is more accurate at predicting exact matches in positive mode at high recall than using MS2Deepscore, the cosine score or the modified cosine score. The threshold for MS2Query, MS2Deepscore, cosine and modified cosine is varied, resulting in different recalls. The random results show the results if random matches would be selected and the optimal results show the performance if the best structural match in the library was selected. **a:** The 'analogues test set' is used with spectra that have no exact match in the library, therefore the best possible match is always an analogue. For MS2Deepscore, cosine score and modified cosine score, library spectra are first filtered on a mass difference of 100 Da. The relationship between recall and average Tanimoto score (chemical similarity) is plotted. For each threshold the average over the Tanimoto scores between the correct molecular structure and the predicted analogues is calculated. **b:** The 'exact match test set' is used, all these test spectra have at least 1 exact structural match in the reference library. For MS2Deepscore and modified cosine score, library spectra are first filtered on a mass difference of 0.25 Da, while MS2Query does not use any pre-filtering on

mass difference, and uses the exact same settings as for the analogue search. The percentage of true positives is given. A match is marked as true positive if the 2D structure is correct. **c:** The same plot as Figure 2.a, but for a model trained on spectra in negative ionization mode. **d:** The same plot as Figure 2.b, but for a model trained on spectra in negative ionization mode.

We rewrote the Material and Methods to match the 20-fold cross validation. The largest change was in the following section:

“MS2Query was designed to search for analogues and exact matches in one run, two different types of test sets were generated to test the performance on these two goals, an “analogue test set” and an “exact matches test set”. Benchmarking for the analogue test set was done using 20-fold cross validation. Meaning that the training dataset was split into 20 test sets. The MS2Deepscore, Spec2Vec and Random Forest model were trained on the remaining 19 sets. The analogue test sets were generated by splitting the unique 2D structures in the library into 20 equal groups, for which all corresponding spectra were selected. Benchmarking for the exact matches test set was also done by creating 20 test sets. These test sets were generated by selecting all unique 2D structures in the library and randomly selecting one spectrum from these, leaving the rest in the library. The 20 sets differ in which spectrum was randomly selected from the unique 2D structures.

The performance of MS2Query was compared to the performance of the modified cosine score, the cosine score and MS2Deepscore. All four methods use a minimal threshold for the spectral similarity score to determine if a library spectrum is a good analogue or exact match. The threshold for each method was varied between 0 and 1, followed by calculating a performance metric and the recall for predictions falling in the selected threshold. The used performance metric for the “analogue test set” was the Tanimoto score between the predicted library molecule and the correct test molecule. The performance metric for the “exact matches test set” was the percentage of true matches, a prediction was considered a true match when it matches the 2D structure of the correct molecule.

For the analogue search using MS2Deepscore and modified cosine score, library spectra were first filtered on a maximum precursor m/z difference of 100 Da. For the benchmarking of the search for exact matches MS2Deepscore and cosine score only library spectra were considered within a mass tolerance of 0.25 Da. For the cosine score the cosine greedy implementation of matchms32 was used to calculate the cosine score, a fragment mass tolerance of 0.05 Da was used.”

We rewrote part of the results section to match the 20-fold change, the largest change was the following section:

“To test the performance of MS2Query models were trained using publicly available mass spectra from GNPS. These spectra were first cleaned and filtered, including cleaning metadata, filtering out spectra with less than 3 peaks and normalizing intensities.

The performance on finding exact matches and finding analogues was tested separately using two different test sets. The test set for searching for exact matches (‘exact match test set’) contains spectra that have at least one spectrum in the library from exactly the same molecule. The test set to test the performance for an analogue search (‘analogues test set’) contains spectra that do not have an exact match to a library spectrum. Thus, for this test set, the best possible match has to be an analogue of the query spectrum.

None of the testing spectra were used for training MS2Deepscore, Spec2Vec and MS2Query to ensure that there is no data leakage between the models. For the exact matches test set the spectra are

spectrum-disjoint, meaning that no spectrum in the test set was used for training of MS2Deepscore, Spec2Vec or MS2Query. The analogue search test set is structure-disjoint, meaning that there were no spectra in the training set that correspond to any of the 2D structures in the test set. The test set is separated on 2D structures, since mass spectra of stereoisomers are often very similar and tandem mass spectrometry often does not have the differentiating power to differentiate between different stereoisomers.

For the analogues test set 20-fold cross validation was performed, the data split was done on all unique 2D structures. For the exact matches test set one test spectrum was randomly selected for each unique 2D structure with at least 2 spectra. This was repeated 20 times, by randomly selecting unique test spectra.

The performance of MS2Query was compared to MS2Deepscore, the cosine score and the modified cosine score. As a metric for the quality of a predicted analogue the average Tanimoto score between the test molecules and predicted analogues is used. The Tanimoto score³⁴ is a metric for chemical similarity between two molecules, based on chemical fingerprints³⁵. For all methods a minimal threshold can be used to vary the percentage of query spectra for which a match is predicted (recall). For all methods, the quality of predictions increases with more stringent thresholds, but the recall decreases. To assess performance for an analogue search, recall is compared to quality of the predictions on the ‘analogues test set’ (Figure 2a). Across all recall values, MS2Query predicts analogues of better quality than comparable search methods relying solely on MS2Deepscore or on the modified cosine score.“

13 b Also, doing splits where we see more challenging data splits would be of interest.

The difficulty of a data split certainly has an impact on the performance of MS2Query and the other benchmarking methods. The rationale of the “analogues test set” was that this is a very difficult data split, since there are no similar 2D structures in the library or in the training data. We think that the combination of these two types of test sets in combination with the 20-fold cross-validation gives a clear overview of the strengths and weaknesses of MS2Query.

14. The modified cosine score is not defined. A section in the supplementary that gives a mathematical formulation would be helpful.

We agree that this should be more clearly defined (being the currently widely-used standard); therefore, we added an explanation to the introduction:

“For an analogue search, it is important to have a spectral similarity score that serves as a good proxy for chemical similarity even if two molecules are not identical. A first improvement made in this direction was the development of the modified cosine score, which in contrast to the cosine score also uses neutral losses for determining spectral similarity^{4, 28}, see Supplementary Information section S8 for more details. This makes the modified cosine score less sensitive to a small chemical modification. However, multiple small chemical modifications can still result in a large decrease in mass spectral similarity, which limits its ability to serve as a proxy for chemical similarity^{19, 29, 30, 31}.”

In the supplementary information we added section 8 which also provides a formula for the modified cosine score:

“S8. Cosine score and Modified Cosine score

The cosine score and modified cosine score are used as reference benchmarking. The cosine score and modified cosine score both calculate spectral similarity by directly comparing spectral peaks. The score is calculated by finding the best possible matches between peaks of two spectra and subsequently calculating a cosine score between the resulting vectors.

Cosine score: when comparing spectrum S to another spectrum S' , a peak p is considered as an eligible match for the cosine score if $mz(p) - mz(p') < t$, where t is the tolerance. The tolerance used in our analysis was 0.05 Da.

Modified cosine score: works as the cosine score, but a peak is also considered as an eligible match if $mz(p) - mz(p') < t$, but also if $mz(p) + M - mz(p') < t$, where $M = PM(S') - PM(S)$. M is the modification mass (difference between two precursor masses) and PM is the precursor mass.”

15. P 5, line 159: “accuracy” is not defined

We changed the term accuracy to quality of predictions to avoid confusion (as per suggestion of reviewer 1). So the sentence was changed to: “To assess performance for an analogue search, recall is compared to quality of the predictions on the ‘analogues test set’ (Figure 2a). As a metric for the quality of a predicted analogue the average Tanimoto score between the test molecules and predicted analogues is used. The Tanimoto score is a metric for chemical similarity between two molecules, based on chemical fingerprints.”

Supplementary material

16. “The model using the feature that first calculates the average MS2Deepscore for each selected library structure, followed by taking the average of the 10 average scores for each library structure performed the best.” So you are using many more spectra for the method that you selected? So averaging many more items is better? Please clarify this issue and potentially remove this section as it seems trivial. It would have been more interesting to see the impact of selecting 5 or 20 items instead of just 10.

This section in the supplementary describes optimization of feature 5; the Average of MS2Deepscore of multiple similar library spectra. The rationale behind this score is that you expect two molecules that are similar in the library to both have a high MS2Deepscore with a good analogue. When comparing a query spectrum against an entire library there is a risk on false positives when searching for analogues. For 10 library structures that are chemically similar we expect to also have a high MS2Deepscore if the library spectra is really a good analogue. By using the average over these 10 structures we expect to filter out false positives.

We tried different methods for calculating the average over all the MS2Deepscores calculated. The same number of spectra were used, since for both methods 10 library structures were used. However, not every library structure has the same number of corresponding spectra in the library. The two approaches of averaging that we tried were simply taking the average over all the compared spectra, meaning that one structure with 200 spectra would contribute a lot more to this score than a structure with only 2 corresponding spectra. The method of averaging that we chose first takes the average per structure, followed by taking the average over the 10 structures.

The number of 10 spectra was selected, since for most structures the 10th structure was still somewhat similar, while for higher number of spectra, the last library structures became too dissimilar. We did try some optimization by inputting the average MS2Deepscore for each of the 10 structures and their Tanimoto scores separately into the random forest model. This did not show improvements to the performance of MS2Query, therefore we expected that changing the number of structures would not have a large effect on the performance of MS2Query.

17. The results for the negative ionization mode are surprising, especially for the exact match. Can you validate the results on the positive mode by training on a similar dataset size and see if the performance drops as it did for negative mode? Can you also pull this result into the main paper ?

We did add the results on the negative mode to Figure 2 in the main paper.

We expect that the difference in performance is partially due to a smaller dataset. However, negative mode spectra also have different characteristics, they are for instance known to have less fragments than positive mode spectra. This means that there is less input data to train on, which might make it more difficult to do good predictions.

We added a sentence in the discussion, to also highlight the different characteristics of negative mode spectra:

“MS2Query performed worse for negative mode mass spectra, which is probably due to the lower number of publicly available mass spectra in negative mode and the fact that negative mode mass spectra contain less mass fragments compared to positive mode mass spectra.”

Minor issues

18. When you used this term, “(Modified) cosine scores”, what is the intention? Are you referring to both cosine and modified cosine scores? Please clarify.

Our intention was indeed to refer to Modified cosine score and cosine score, and we agree that this was confusing. This was changed at all occurrences throughout the text.

19. “the computation time to determine the fragmentation trees also increases manifold.” This sentence can be made more precise regarding the runtime for fragmentation trees

We added our review paper that discusses this increase in runtime. (de Jonge, N.F., Mildau, K., Meijer, D., Louwen, J.R., Bueschl, C., Huber, F. and van der Hooft, J.J. (2022) Good Practices and Recommendations for Using and Benchmarking Computational Metabolomics Metabolite Annotation Tools).

Our review paper mentions:

“Within SIRIUS, the number of possible fragmentation trees, and therefore number of predicted molecular formulas that are computed increase exponentially with higher masses. This leads to reduced performance in molecular formula determination for masses higher than 500 Da (Böcker, Dührkop 2016; Böcker et al. 2008). Using the recently developed ZODIAC method, this issue is partially resolved by reranking the lists of molecular formula candidates in larger MS/MS datasets leading to substantially lower error rates for molecular formula assignment (Ludwig et al. 2020). Besides this, fragmentation tree computation is a NP-hard problem and therefore puts a time constraint on the performance of SIRIUS. Spectra with masses above 850 Da are therefore not able to be computed within realistic timescales. To illustrate, the full Actinomycetes (Salinispora/Streptomyces) dataset used in MolNetEnhancer takes over 4 weeks to compute using the SIRIUS workflow, compared to around 24 hours when using the same computational resources and a precursor mass cut-off of 850 Da.”

20. “The percentage of true positives is given, a match is marked as true positive if the first 14 characters of the InChiKeys are identical.” Incorrect punctuation. This is not a good English sentence.

Changed to: “The percentage of true positives is given. A match is marked as true positive if the first 14 characters of the InChiKeys are identical.”

21. “These methods perform especially well at predicting chemical similarity for molecules that are similar, but are chemically not exact matches. This makes these scores very suitable for an analogue search.” The last sentence is not clear. Which scores?

Changed to: “This makes Spec2Vec and MS2Deepscore very suitable for an analogue search.”

22. “your query spectra” – may just called it “the query spectra”? this appears in multiple places in the text.

Rephrased at both instances.

23. “For all other benchmarking only the positive mode spectra are used”. What other benchmarking is being referred to? Please rewrite to clarify.

We did refer to the benchmarking in Figure 2, 3, S1, S2, S3, S4, S5 in the previous manuscript version. This section has been rewritten to describe the benchmarking workflow for the k-fold cross-validation. In the supplementary figures we now clearly mention for each figure if it was positive or negative mode.

Add DATA AND CODE AVAILABILITY

A code availability statement was added, and the Data Availability section was updated. The data availability and code availability section can be found below.

Data availability

The models and spectra files used for the case studies can be downloaded from <https://zenodo.org/record/6124553>. For the k-fold cross validation the raw results, the raw data and the data splits can be downloaded from <https://zenodo.org/record/7427094>.

The mass spectrometry data used for the case studies were deposited in the MassIVE repository (accession number MSV000089648 and MSV000090642).

Code availability

MS2Query is available as an easily installable Python library running on Python 3.7 and 3.8 on Windows, Linux and MacOS. Source code and installation instructions can be found on Github (<https://github.com/iomega/ms2query>). The case study results were obtained using version 0.3.2 and the k-fold-cross validation results were obtained using version 0.5.6.

REVIEWERS' COMMENTS

Reviewer #3 (Remarks to the Author):

Thank you for addressing the concerns that were raised during the review.

There remain the following concerns:

Regarding 7C.

Thank you for responding to us. However, the issue of the performance of MS2Query when recall is high needs to be clarified in the manuscript. Please comment on this limitation/issue in the paper.

Regarding 13.

Moving to cross validation is a good idea. However, there is no explanation of why the choice was "20", which is odd considering that 5 or 10-fold cross validation is the norm. Please explain in the paper this choice. Please be clear if there is an issue related to the size of the dataset.

Regarding 16.

Please write a paragraph and experiments in the paper to support your choices. This will highlight that your choices are based on experimental observations and will increase understanding of the choices made to tune the heuristic.

Wageningen, the Netherlands, March 3, 2023

We thank the reviewers and editor for studying our revised manuscript and we were happy to read that you appreciated our previous efforts in further improving our benchmarking analyses and clarifying methodological aspects of MS2Query.

Please find below the point-by-point response to the remaining remarks of Reviewer 3, where our response is written in blue. We have also checked the Editorial checklist - our responses were logged in the document itself that is uploaded separately.

Regarding 7C.

Thank you for responding to us. However, the issue of the performance of MS2Query when recall is high needs to be clarified in the manuscript. Please comment on this limitation/issue in the paper.

We agree that it is good to explain this in the main text as well, and in the Results section we added the following sentences:

“At a high recall, the observed increase in performance is smaller, which suggests that the main added value of MS2Query is a better removal of bad matches as compared to the other methods. This demonstrates the importance of using a sufficiently high threshold for the MS2Query score.”

Regarding 13.

Moving to cross validation is a good idea. However, there is no explanation of why the choice was “20”, which is odd considering that 5 or 10--fold cross validation is the norm. Please explain in the paper this choice. Please be clear if there is an issue related to the size of the dataset.

Choosing 5 or 10-fold cross-validation is indeed a common choice. We chose for 20-fold cross validation, since with this value, the size of the test and training sets is similar to the benchmarking in the first version of the manuscript. This seemed like the right balance between computation cost and ensuring a large enough training set.

To also explain this in the paper, we added the following sentence in the Results section:

“20-fold cross-validation was chosen to ensure that a large-enough training set was used to not compromise on overall model performance.”

Regarding 16.

Please write a paragraph and experiments in the paper to support your choices. This will highlight that your choices are based on experimental observations and will increase understanding of the choices made to tune the heuristic.

We agree with the reviewer and, actually, we think that the requested information is present in Supplementary Note 3, which does contain multiple sections about the experimentation

we did and the resulting model performance that were used to make our choices. However, to provide the interested reader with a more direct pointer to this information, we added to the manuscript in the Methods section:

“Multiple variations of the implementation of this feature were tested and the best performing implementation was selected. These other implementations used weighting based on the Tanimoto score, or weighting the MS2Deepscore for each spectrum equally instead of using the average MS2Deepscore per library structure. These other implementations and their performance are described in more detail in Supplementary Note 3.”

On behalf of all the authors,

Niek de Jonge, Florian Huber, and Justin van der Hooft